# Diffusion-NPO: Negative Preference Optimization for Better Preference Aligned Generation of Diffusion Models

**Fu-Yun Wang**[1]  **Yunhao Shui**[2]  **Jingtan Piao**[1]  **Keqiang Sun**[1]  **Hongsheng Li**[1,3]

[1] MMLab, CUHK, Hong Kong    [2] Shanghai Jiang Tong University, Shanghai    [3] CPII under InnoHK, Hong Kong

`fywang@link.cuhk.edu.hk, xilanhua12138@sjtu.edu.cn, 1155116308@link.cuhk.edu.hk`
`kqsun@link.cuhk.edu.hk, hsli@ee.cuhk.edu.hk`

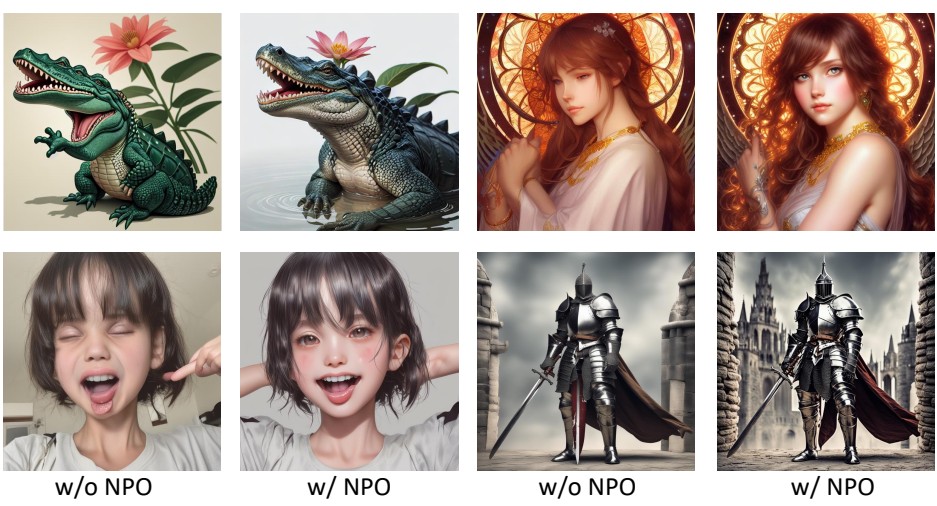

| w/o NPO | w/ NPO | w/o NPO | w/ NPO |

Figure 1: Diffusion-NPO enhances high-frequency details, color and lighting, and low-frequency structures in images by aligning human's negative preference.

## Abstract

Diffusion models have made substantial advances in image generation, yet models trained on large, unfiltered datasets often yield outputs misaligned with human preferences. Numerous methods have been proposed to fine-tune pre-trained diffusion models, achieving notable improvements in aligning generated outputs with human preferences. However, we argue that existing preference alignment methods neglect the critical role of handling unconditional/negative-conditional outputs, leading to a diminished capacity to avoid generating undesirable outcomes. This oversight limits the efficacy of classifier-free guidance (CFG), which relies on the contrast between conditional generation and unconditional/negative-conditional generation to optimize output quality. In response, we propose a straightforward but versatile effective approach that involves training a model specifically attuned to negative preferences. This method does not require new training strategies or datasets but rather involves minor modifications to existing techniques. Our approach integrates seamlessly with models such as SD1.5, SDXL, video diffusion models and models that have undergone preference optimization, consistently enhancing their alignment with human preferences.

## 1 Introduction

Diffusion models have made significant strides in image/video generation (Ho et al., 2020; Rombach et al., 2022; Podell et al., 2023; Dhariwal & Nichol, 2021; Singer et al., 2022; Shi et al., 2024; Wang

et al., 2024f;c;e;b; Liang et al., 2024a; Liu et al., 2022; Peebles & Xie, 2023; Karras et al., 2022; Ke et al., 2024; Yin et al., 2024). However, diffusion models trained on massive unfiltered image-text pairs (Schuhmann, 2022; Sun et al., 2024) often generate results that do not align with human preferences. To address this issue, many methods (Wu et al., 2023; 2024) have been proposed to align diffusion models with human preferences, aiming to drive the generation to better match what users desire.

Human preference alignment methods typically require the prior collection of a human preference dataset, such as Pick-a-pic (Kirstain et al., 2023). The standard procedure involves gathering pairs of images generated from the same prompt and annotating them according to human preferences. Rather than assigning direct scores, these preferences are usually ranked in order. This ranking is then utilized to train a scoring/reward model for text-image pairs using a contrastive loss function (Ouyang et al., 2022). To explore this topic in depth, we first review existing approaches for aligning diffusion models with human preferences. In general, current methods can be categorized into three types:

a) **Differentiable Reward (DR)**: These approaches directly feed multi-step generated images into a pretrained reward model, updating the diffusion models through gradient backpropagation (Xu et al., 2024; Prabhudesai et al., 2024; Zhang et al., 2024b; Wu et al., 2023; 2024). While simple and direct, these methods are prone to reward leakage (Zhang et al., 2024b).

b) **Reinforcement Learning (RL)**: In these approaches, the denoising process of diffusion models is formulated as an equivalent Markov decision process (MDP) (Puterman, 2014). PPO (Schulman et al., 2017) and its variants are typically adopted for preference optimization. Images are generated and evaluated online based on the reward feedback, aiming to increase the probability of generating high-reward images. These approaches employ SDE solvers to achieve stochastic sampling and importance sampling (Sutton, 2018).

c) **Direct Preference Optimization (DPO)**: These approaches simplify the reinforcement learning training objective into a straightforward simulation-free training objective (Rafailov et al., 2024; Wallace et al., 2024). They do not require training reward models, nor do they need online generation and sampling; instead, they only require fine-tuning on pre-collected paired preference data. Although simple, these approaches often underperform reinforcement learning-based methods, especially for out-of-distribution inputs.

Despite previous efforts to make models generate human-aligned images, we raise an important question: *How can a model know to avoid generating poor images if it only knows how to generate good ones without understanding what is bad?*

We identify a crucial oversight in current diffusion model preference alignment efforts: most diffusion generation rely heavily on the classifier-free guidance (CFG) (Ho & Salimans, 2022; Karras et al., 2024; Shen et al., 2024; Ahn et al., 2024; Wang et al., 2024a;d). CFG requires the model to simultaneously compute outputs under both conditional inputs and negative-conditional/unconditional inputs at each denoising step, then linearly combine these outputs to bias the final prediction towards the conditional inputs and away from the negative-conditional inputs. Ideally, we expect the model's output under the conditional inputs to align closely with human preferences, while the output under the negative-conditional inputs should diverge from human preferences to maximize preference alignment. However, previous works focus exclusively on training models to generate outputs that align with human preferences, without considering the equally important task of teaching models to recognize and avoid generating outputs that humans do not favor. This oversight limits the effectiveness of existing alignment strategies, particularly in scenarios where distinguishing between preferred and non-preferred outputs is crucial.

To address this issue, we propose **N**egative **P**reference **O**ptimization (NPO): training an additional model that is aligned with preferences opposite to human. Importantly, our crucial insight is that *training such a negative preference aligned model requires no new training strategies or datasets, only minor modifications to existing methods.* 1) Approaches like differential reward and reinforcement learning, all need a reward model for training. We simply multiply the output of reward model by $-1$, which allows us to train a negative preference model using the same approaches. 2) For DPO-based methods, we reverse the order of the preferred image pairs. Notably, during the training of the reward model applied for differential reward and reinforcement learning approaches, the im-

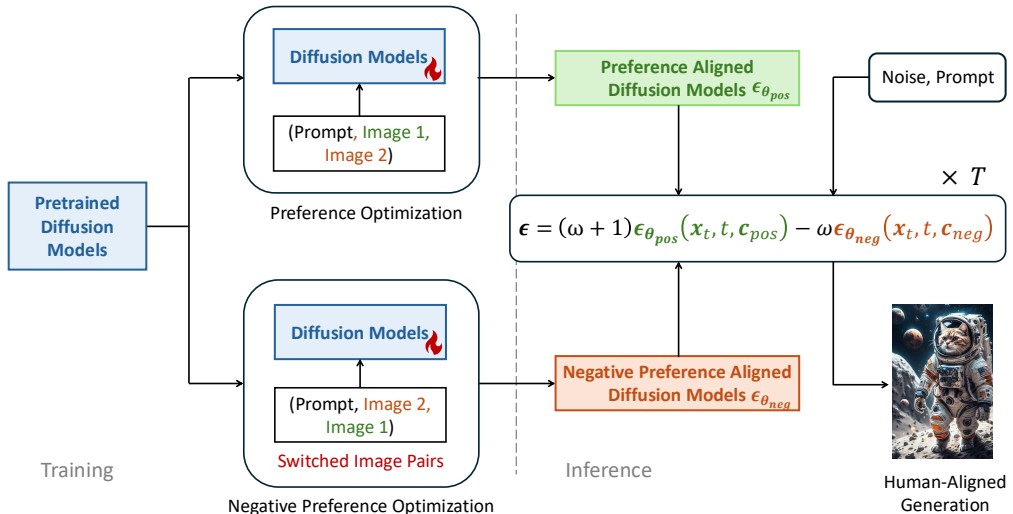

Figure 2: High-level overview of negative preference optimization (NPO). (Training) NPO needs no new training strategies and datasets. NPO training can be achieved through switching preference image pairs with existing preference optimization methods. (Inference) NPO trained models serve as the unconditional/negative-conditional predictors in the classifier-free guidance.

age order can also be reversed to train the reward model. Therefore, in essence, all strategies can be perceived as reversing the order of image pairs in the collected preference data by adapting the same training procedure. Fig. 2 provides an overview of our method.

We validate the effectiveness of NPO on text-to-image generation with SD1.5 (Rombach et al., 2022) and SDXL (Podell et al., 2023) and text-to-video generation with VideoCrafter2 (Chen et al., 2024). Our model can be used in a plug-and-play manner with these baseline models and their various preference-optimized versions, consistently improving generation quality. Fig. 3 shows our comparative results. We evaluate our method using the widely adopted Pick-a-pic validation set, scoring with metrics including HPSv2, ImageReward, PickScore, and LAION-Aesthetic. Our approach significantly improves performance across all metrics.

## 2 UNDERSTANDING CLASSIFIER-FREE GUIDANCE

**Preliminary of CFG.** CFG has became a necessary and important technique for improving generation quality and text alignment of diffusion models. For convenience, we focus our discussion on the general formal of diffusion models, *i.e.*, $\boldsymbol{x}_t = \alpha_t \boldsymbol{x}_0 + \sigma_t \boldsymbol{\epsilon}$ (Kingma et al., 2021). Suppose we learn a score estimator from a epsilon prediction neural network $\boldsymbol{\epsilon}_{\boldsymbol{\theta}}(\boldsymbol{x}_t, \boldsymbol{c}, t)$, and we have $\nabla_{\boldsymbol{x}_t} \log \mathbb{P}_{\boldsymbol{\theta}}(\boldsymbol{x}_t | \boldsymbol{c}; t) = -\frac{\boldsymbol{\epsilon}_{\boldsymbol{\theta}}(\boldsymbol{x}_t, t)}{\sigma_t}$. The sample prediction at timestep $t$ of the score estimator is formulated as

$$\hat{\boldsymbol{x}}_0 = \frac{1}{\alpha_t}(\boldsymbol{x}_t + \sigma^2 \nabla_{\boldsymbol{x}_t} \log \mathbb{P}_{\boldsymbol{\theta}}(\boldsymbol{x}_t | \boldsymbol{c}; t)). \tag{1}$$

Applying the CFG is equivalent to add an additional score term (Karras et al., 2024), that is, we replace $\nabla_{\boldsymbol{x}_t} \log \mathbb{P}_{\boldsymbol{\theta}}(\boldsymbol{x}_t | \boldsymbol{c}; t)$ in Eq. 1 with the following term,

$$\nabla_{\boldsymbol{x}_t} \log \mathbb{P}_{\boldsymbol{\theta}}(\boldsymbol{x}_t | \boldsymbol{c}; t) + \nabla_{\boldsymbol{x}_t} \log \left[ \frac{\mathbb{P}_{\boldsymbol{\theta}}(\boldsymbol{x}_t | \boldsymbol{c}; t)}{\mathbb{P}_{\boldsymbol{\theta}}(\boldsymbol{x}_t | \boldsymbol{c}'; t)} \right]^{\omega}, \tag{2}$$

where $\omega$ is to control the strength of CFG, $\boldsymbol{c}$ and $\boldsymbol{c}'$ are conditional and unconditional/negative-conditional inputs, respectively. It is apparent that the generation will be pushed to high probability region of $\mathbb{P}_{\boldsymbol{\theta}}(\boldsymbol{x}_t | \boldsymbol{c}; t)$ and relatively low probability region of $\mathbb{P}_{\boldsymbol{\theta}}(\boldsymbol{x}_t | \boldsymbol{c}'; t)$. Write the above equation into the epsilon format, and then we have

$$\boldsymbol{\epsilon}_{\boldsymbol{\theta}}^{\omega} = (\omega + 1)\boldsymbol{\epsilon}_{\boldsymbol{\theta}}(\boldsymbol{x}_t, \boldsymbol{c}, t) - \omega \boldsymbol{\epsilon}_{\boldsymbol{\theta}}(\boldsymbol{x}_t, \boldsymbol{c}', t). \tag{3}$$

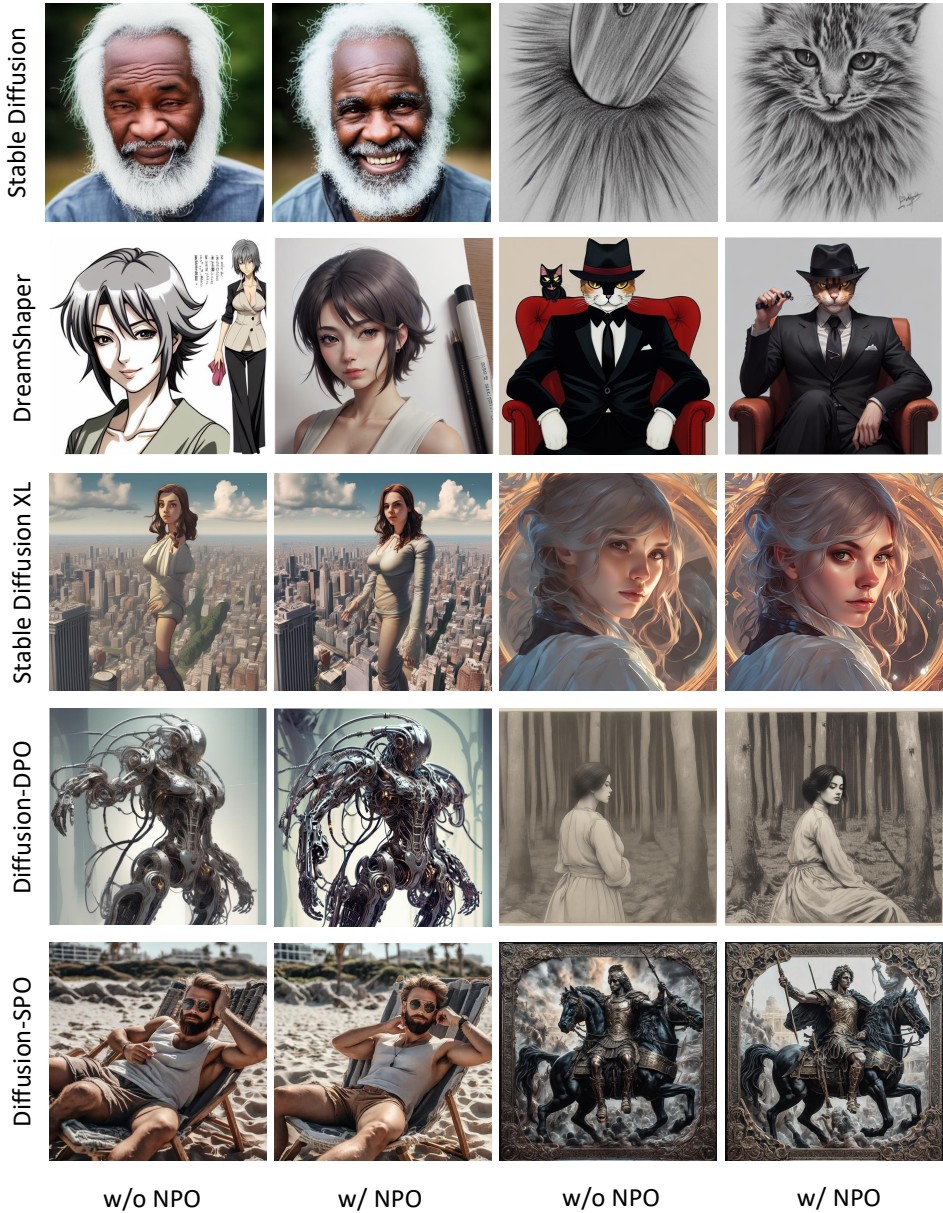

Figure 3: NPO works as a plug-and-play inference enhancement strategy. It can be easily combined with base diffusion models and preference optimized diffusion models for better human preference aligned generation. Zoom out for better comparison in details.

**Motivating example.** To maximize human preference alignment, in Eq. 3, the green component should guide the generated results to closely match human preferences, while the orange component should direct the results away from undesired outcomes. However, most preference optimization methods focus exclusively on optimizing the green component, neglecting the orange component and thereby weakening its impact. To illustrate this point, we setup a motivating experiment to investigate the influence of the orange component. We employ two baselines:

1. We use the DPO-optimized SD1.5 (Wallace et al., 2024; Rombach et al., 2022) for both the green component and the orange component.

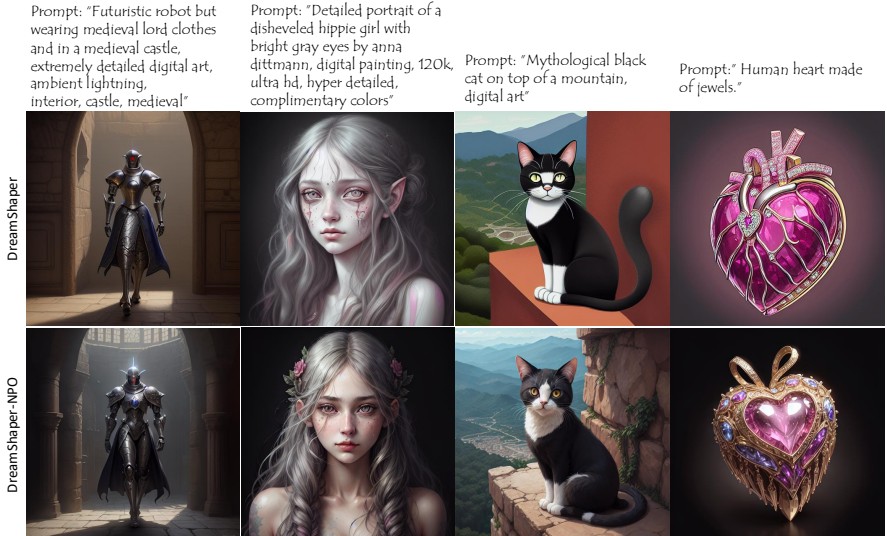

Prompt: "Futuristic robot but wearing medieval lord clothes and in a medieval castle, extremely detailed digital art, ambient lightning, interior, castle, medieval"

Prompt: "Detailed portrait of a disheveled hippie girl with bright gray eyes by anna dittmann, digital painting, 120k, ultra hd, hyper detailed, complimentary colors"

Prompt: "Mythological black cat on top of a mountain, digital art"

Prompt:" Human heart made of jewels."

Figure 4: Plug-and-play NPO on DreamShaper. NPO not only works on the base Stable Diffusion and its preference optimized variants, but also works on improving customized model finetuned on high-quality data.

2. We use the DPO-optimized SD1.5 for the green component and the model merged from weights of DPO-optimized SD1.5 ($0.6\times$) and original SD1.5 ($0.4\times$) for the orange component, generating results with the same seed.

We compare the generated images from the two baslines one by one, score them using HPSv2, ImageReward, PickScore, and LAION-Aesthetic, and calculate their win probabilities. The results are shown in Fig. 5. We can observe a significant improvement in human preference compared to only using the DPO-optimized model.

**Analysis: the weight merge is an approximated NPO.** What is the meaning of the weight merged model? Suppose the weight of original SD is $\boldsymbol{\theta}$, and then the DPO-optimized model weight can be denoted as $\boldsymbol{\theta} + \boldsymbol{\eta}$ since it is further trained from the original weight $\boldsymbol{\theta}$. The merged model weight is

$$\gamma(\boldsymbol{\theta}+\boldsymbol{\eta})+(1-\gamma)\boldsymbol{\theta} = \boldsymbol{\theta}+\gamma\boldsymbol{\eta} = \boldsymbol{\theta}+\boldsymbol{\eta}+(1-\gamma)(-\boldsymbol{\eta}), \quad (4)$$

where $\gamma \in [0, 1]$ is the merge factor. We can observe that after weight fusion, the weight offset obtained through DPO optimization $\boldsymbol{\eta}$ has decreased. Consequently, the DPO weight offset $\boldsymbol{\eta}$ has a weaker impact on the generated results, enabling the model to output results that are more contrary to human preferences. Replace $(1 - \gamma)(-\boldsymbol{\eta})$ with $\boldsymbol{\delta}$, and then the weight can be represented as

$$\boldsymbol{\theta}' = \boldsymbol{\theta} + \boldsymbol{\eta} + \boldsymbol{\delta}. \quad (5)$$

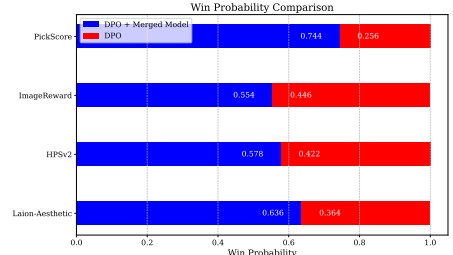

Figure 5: Motivating example results of Section 2. Applying merged model as the orange component (*i.e.*, prediction for unconditional/negative-conditional inputs) effectively improves the human preference alignment.

The above equation decomposes the weight applied for negative-conditional predictions into three parts: the original model weight $\boldsymbol{\theta}$, the preference alignment weight offset (direction) $\boldsymbol{\eta}$, a weight offset opposite to the preference alignment $\boldsymbol{\delta}$. Our paper aims to train a suitable $\boldsymbol{\delta}$ and investigate its properties. Note that, once the $\boldsymbol{\eta}$ and $\boldsymbol{\delta}$ are obtained, we can also flexibly change the influence of each weight offset by multiply simple scale factors. That is,

$$\boldsymbol{\theta}' = \boldsymbol{\theta} + \alpha\boldsymbol{\eta} + \beta\boldsymbol{\delta}, \quad (6)$$

where $\alpha, \beta \in [0, 1]$.

Table 1: Quantitative performance comparison with stable diffusion v1-5 based models. * means the metrics are copied from SPO papers. Other metrics are tested with official weights.

| Method | PickScore | HPSv2 | ImageReward | Aesthetic |
|---|---|---|---|---|
| SD-1.5 | 20.75 | 26.84 | 0.1064 | 5.539 |
| *DDPO | 21.06 | 24.91 | 0.0817 | 5.591 |
| *D3PO | 20.76 | 23.97 | -0.1235 | 5.527 |
| Diff.-DPO | 20.98 | 25.05 | 0.1115 | 5.505 |
| Diff.-SPO | 21.41 | 26.85 | 0.1738 | 5.946 |
| SD-1.5 + NPO | 21.26 | 27.36 | 0.2028 | 5.667 |
| Diff.-SPO + NPO | **21.65** | 27.09 | 0.1939 | **5.999** |
| Diff.-DPO + NPO (reg= 500) | 21.58 | **27.60** | 0.3101 | 5.762 |
| Diff.-DPO + NPO (reg= 1000) | 21.43 | 27.36 | **0.3472** | 5.773 |
| DreamShaper | 21.96 | 27.97 | 0.7131 | 6.085 |
| DreamShaper + NPO ($\alpha = 1.0$) | 22.38 | **28.31** | **0.7396** | 6.169 |
| DreamShaper + NPO ($\alpha = 0.6$) | **22.46** | 28.08 | 0.6626 | **6.496** |

## 3 NEGATIVE PREFERENCE OPTIMIZATION

Previous approaches primarily focus on training single model weight that aligns with human preferences. However, these methods often overlook the importance of unconditional outputs of classifier-free guidance in the diffusion generation process. Our approach seeks to train a weight offset $\boldsymbol{\delta}$ that opposes human preferences to fulfill the role of unconditional outputs. By integrating this offset with the base model's weights, it functions as a predictor for unconditional inputs, thereby reducing the likelihood of generating outputs that conflict with human preference. The important motivation for negative preference optimization is that a preference-aligned model should not only learn to generate desirable outcomes but also understand what constitutes undesirable ones. This dual understanding is crucial for maximizing preference alignment while minimizing the occurrence of unwanted results.

### 3.1 TRAINING WITH NPO

An important insight in our work is that achieving negative preference optimization does not require new datasets, reward models, or even new training strategies. Standard preference optimization methods can be directly applied to negative preference optimization.

For methods based on reinforcement learning and differential rewards, which typically rely on a reward model $R(\mathbf{x}, \boldsymbol{c}) \in [0, 1]$ (can be easily scaled if not in this interval). This reward model can be transformed into the form required for negative preference optimization as follows:

$$R_{\text{NPO}}(\mathbf{x}, \boldsymbol{c}) = 1 - R(\mathbf{x}, \boldsymbol{c}). \tag{7}$$

Fo methods that utilize reward models, we can simply substitute the original $R(\mathbf{x}, \boldsymbol{c})$ in the algorithm with $R_{\text{NPO}}(\mathbf{x}, \boldsymbol{c})$.

For methods that train on preference pairs $r = (\mathbf{x}_0, \mathbf{x}_1, \boldsymbol{c})$, where $\mathbf{x}_0$ is less preferred and $\mathbf{x}_1$ is more preferred by humans, and $\boldsymbol{c}$ is the conditional information used for generation (indicating both images are generated from the same $\boldsymbol{c}$), converting this to a negative preference optimization algorithm requires simply reversing the order of the preference pair:

$$r_{\text{NPO}} = (\mathbf{x}_1, \mathbf{x}_0, \boldsymbol{c}). \tag{8}$$

Beyond the fundamental implementation of negative preference optimization outlined above, it is important to recognize that many preference optimization methods may use CFG during training for sample collection, probability calculation, and gradient backpropagation. NPO can naturally extend to these methods as well. Although these methods might apply CFG during training to bridge the gap between training and inference, they train only a single weight offset, overlooking the fact that the conditional and unconditional (or negative-conditional) outputs in CFG have different optimization objectives (i.e., preference-aligned and negative preference-aligned). This could result in a weight offset that is a compromise between the two opposite objectives, failing to fully achieve preference alignment. We propose to optimize two distinct weight offsets simultaneously.

Table 2: Quantitative performance comparison with stable diffusion XL based models. All metrics are tested with official weights.

| Method | PickScore | HPSv2 | ImageReward | Aesthetic |
|---|---|---|---|---|
| SDXL | 22.06 | 27.89 | 0.6246 | 6.114 |
| Diff.-DPO | 22.57 | 28.58 | 0.8767 | 6.099 |
| Diff.-SPO | 22.97 | 28.58 | 1.032 | 6.348 |
| SDXL + NPO | 22.32 | 28.11 | 0.6831 | 6.136 |
| Diff.-DPO + NPO | 22.69 | **28.78** | 0.9210 | 6.112 |
| Diff.-SPO + NPO | **23.08** | 28.70 | **1.047** | **6.438** |

## 3.2 Inference with NPO

Let $\boldsymbol{\theta}$ denote the base model weight, $\boldsymbol{\eta}$ the weight offset after preference optimization, and $\boldsymbol{\delta}$ the weight offset after negative preference optimization. A straightforward strategy is to define $\boldsymbol{\theta}_{pos} = \boldsymbol{\theta} + \boldsymbol{\eta}$ and $\boldsymbol{\theta}_{neg} = \boldsymbol{\theta} + \boldsymbol{\delta}$, and then apply classifier-free guidance as follows:

$$\epsilon_{\boldsymbol{\theta}}^{\omega} = (\omega + 1)\epsilon_{\boldsymbol{\theta}_{pos}}(\boldsymbol{x}_t, \boldsymbol{c}, t) - \omega\epsilon_{\boldsymbol{\theta}_{neg}}(\boldsymbol{x}_t, \boldsymbol{c}', t). \quad (9)$$

However, this approach often results in a significant output discrepancy between $\boldsymbol{\theta}_{pos}$ and $\boldsymbol{\theta}_{neg}$. The outputs from classifier-free guidance should maintain a necessary level of correlation; for example, if two Gaussian noises are completely independent, the variance from the operation above would change from 1 to $2\omega^2 + 2\omega + 1$. We find that it is typically necessary to incorporate the positive weight offset into the negative weights, such that:

$$\boldsymbol{\theta}_{neg} = \boldsymbol{\theta} + \alpha\boldsymbol{\eta} + \beta\boldsymbol{\delta}, \quad \alpha, \beta \in [0, 1] \quad (10)$$

which aligns with our earlier motivating example and analysis.

## 4 Experiments

### 4.1 Validation Setup

To validate the effectiveness and versatility of our approach, we test it on three baseline methods:

a) **Diffusion-DPO.** Diffusion-DPO (Wallace et al., 2024) is the first method to incorporate the Direct Preference Optimization (DPO) approach into diffusion training. It introduces a simulation-free and reward model-free training strategy that enables direct training with preference pairs. The effectiveness of this method has been validated on popular text-to-image models, such as the 0.9B Stable Diffusion v1-5 and the 3B Stable Diffusion XL.

b) **Diffusion-SPO.** Diffusion-SPO (Liang et al., 2024b) combines the DPO approach with reinforcement learning. It involves online sample generation, stochastic solvers, and probability calculations, while utilizing the DPO optimization objective for training. This method requires a reward model to score preferences for generated images online. Its effectiveness has also been demonstrated on the 0.9B Stable Diffusion v1-5 and the 3B Stable Diffusion XL for text-to-image generation.

c) **VADER.** VADER (Prabhudesai et al., 2024) is a differential reward-based approach that has shown effectiveness in text-to-video generation, significantly enhancing the aesthetic quality of generated videos from raw models.

Therefore, our validation baselines include differential reward, reinforcement learning, and direct preference optimization (the three typical kinds of methods we mentioned), covering both text-to-image and text-to-video tasks. We believe our validation is sufficiently convincing to demonstrate the effectiveness of our approach. Unless otherwise specified, we use the default training and inference configurations for all the aforementioned methods, including training data, number of training iterations, CFG strength, etc.

### 4.2 Comparison

**Quantitative comparison.** For text-to-image generation, we conduct the quantitative evaluation of our method by following previous work and using the Pick-a-pic 'test_unique' split as the test-

Prompt: "A person playing a guitar by a campfire under a starry sky."

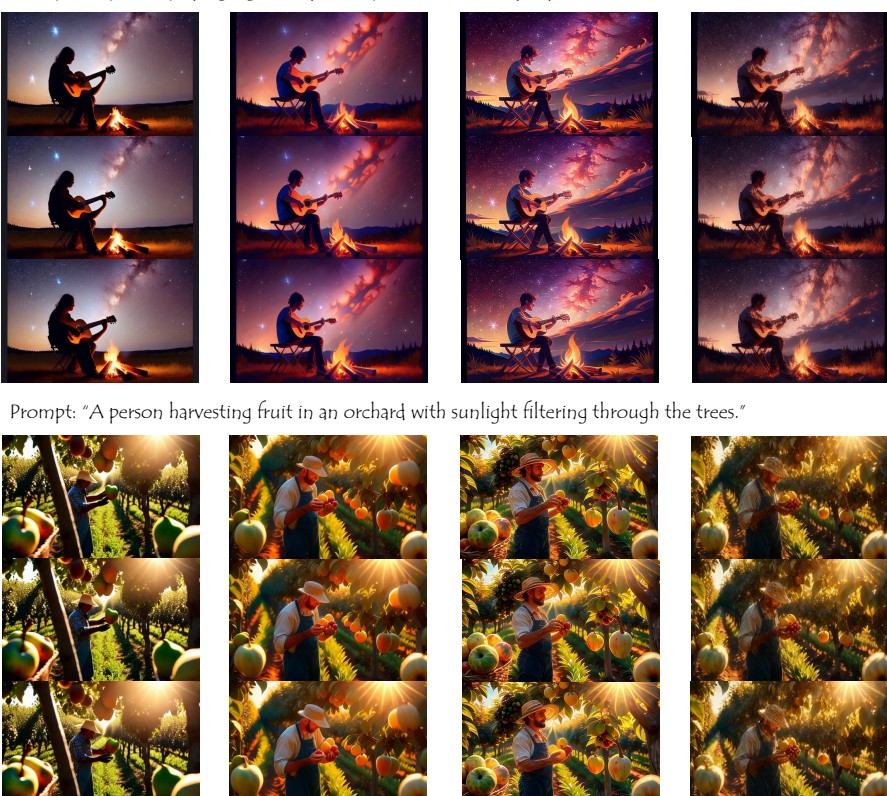

Prompt: "A person harvesting fruit in an orchard with sunlight filtering through the trees."

| VideoCrafter2 | VADER | VADER + NPO (HPSv2) | VADER + NPO (PickScore) |

Figure 6: Video comparison. The videos are trained using 12 frames. For better visualization, we sample one key frame from every four frames..

ing benchmark (Kirstain et al., 2023). We employ PickScore (Kirstain et al., 2023), HPSv2 (Wu et al., 2023), ImageReward (Xu et al., 2024), and Laion-Aesthetic (Schuhmann, 2022) as evaluation metrics. The results of the quantitative evaluation are summarized in Tables 1 and 2. The tables demonstrate that NPO, when combined with the base model and its preference-optimized versions, consistently enhances the aesthetic quality of the generated results. In addition to reporting the average scores, as illustrated in Fig. 7, we calculate the proportion of samples generated with the same prompt that achieve a higher preference score. The results generated using NPO significantly outperform those without NPO. For text-to-video generation, we compare four baselines: VideoCrafter2, VADER, VADER + NPO (HPSv2), and VADER + NPO (PickScore). Among these, VADER + NPO (HPSv2) is optimized using both HPSv2 and Laion-Aesthetic as reward models, while VADER + NPO (PickScore) is optimized using PickScore as the reward model. We train the models using animal-related prompts, as was done with VADER, and evaluate on unseen animal-related prompts (same domain) and additional human prompts (out domain). The results, presented in Table 3, reveal that VADER + NPO (HPSv2) shows significant improvements across all four metrics, particularly in the HPS and Laion-Aesthetic metrics. VADER + NPO (PickScore) demonstrates greater improvement in the PickScore metric and, on animal-related prompts, even achieves better HPSv2 performance than VADER + NPO (HPSv2).

**Qualitative comparison.** Fig. 3, Fig. 4, Fig. 6, Fig. 11, Fig. 12, Fig. 13, Fig. 14 and Fig. 15, present a comparison of results generated with and without NPO across various scenarios. We observe that NPO significantly enhances high-frequency details, color and lighting, and low-frequency structures in images, consistently improving human preference scores.

**User preference.** We assess the generation quality in three specific areas: Color and Lighting, High-Frequency Details, and Low-Frequency Composition. For Color and Lighting, users evaluate whether the generated images display natural and visually pleasing color schemes and lightings. For

Table 3: Quantitative performance comparison on text-to-video generation. All metrics are tested with official weights. Avg means the average score. Win means the average winning ratio to other methods. HPSv2 means we apply both aesthetic predictor and HPSv2 for training. PickScore means we apply PickScore for training.

| Method | Aesthetic | | HPSv2 | | ImageReward | | PickScore | |
|---|---|---|---|---|---|---|---|---|
| | Avg | Win | Avg | Win | Avg | Win | Avg | Win |
| Animal | | | | | | | | |
| VideoCrafter2 | 5.527 | 0.00% | 29.65 | 2.08% | 1.368 | 30.73% | 22.44 | 16.81 % |
| VADER | 6.154 | 55.21% | 32.24 | 46.88% | 1.486 | 58.33% | 22.97 | 34.23% |
| VADER + NPO (PickScore) | 6.110 | 50.00% | **32.81** | **82.81%** | 1.463 | 53.65% | **24.16** | **98.44%** |
| VADER + NPO (HPSv2) | **6.379** | **94.79%** | 32.52 | 68.23% | **1.492** | **59.96%** | 23.14 | 50.52% |
| Human | | | | | | | | |
| VideoCrafter2 | 5.726 | 1.04% | 27.92 | 10.27% | 0.9583 | 33.33% | 22.41 | 27.75% |
| VADER | 6.462 | 61.46% | 29.74 | 51.71% | 1.102 | 46.35% | 22.55 | 34.23% |
| VADER + NPO (PickScore) | 6.244 | 39.58% | 29.58 | 51.71% | 1.086 | 55.73% | **23.35** | **89.06%** |
| VADER + NPO (HPSv2) | **6.855** | **97.92%** | **30.76** | **86.98%** | **1.164** | **64.58%** | 22.71 | 48.29% |

(a) Stable Diffusion v1-5

(b) Diffusion-DPO

(c) Diffusion-SPO

(d) Stable Diffusion XL

(e) Diffusion-DPO-SDXL

(f) Diffusion-SPO-SDXL

Figure 7: Quantitative winning ratios.

High-Frequency Details, users assess the level of detail in textures and the sharpness of fine features, such as edges and small-scale elements. For Low-Frequency Composition, users examine the overall structure and balance of the images. We conduct the user study using the prompts from Pickapic 'validation_unique', with different models generating images based on the same random seed. Users evaluate the models on the three aspects mentioned above and have three choices: "No Preference" (draw), "NPO is better," or "NPO is worse." We distribute questionnaires to 15 volunteers online, with each questionnaire containing 50 pairs of generated images (randomly sampled from SDXL, SD15, and DreamShaper). A total of 750 votes are collected. The final results are presented in the Fig. 8. The user study indicates that NPO significantly enhances high-frequency details in the generated outputs, while also producing colors and lighting that align more closely with human preferences. Additionally, NPO can improve the compositional structure of the generated images to some extent.

**Hyper-parameter sensitivity analysis.** Negative preference optimization involves a crucial trade-off regarding how much the unconditional/negative-conditional outputs deviate from the conditional outputs.f the deviation is too small, the optimization becomes ineffective; if too large, it may result in blurred or unnatural images. During training, this trade-off is managed by controlling how much the weights diverge from those of the base model. Preference optimization methods, such as Diffusion-DPO, often use a regularization factor (Beta) to control the degree of deviation. For inference, this trade-off is determined by how much of the positive weight offset $\eta$ is incorporated into the negative weight $\alpha$. We use the DPO algorithm to train NPO and systematically test this trade-off. Fig. 9 and Fig. 10 show examples of generated images with different parameter settings and the corresponding changes in quantitative metrics. The results indicate that choosing suitable parameters can significantly improve performance.

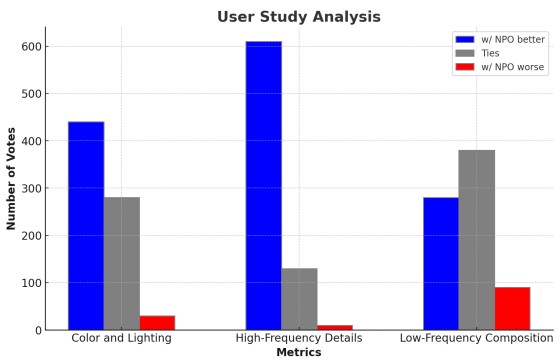

Figure 8: User study analysis.

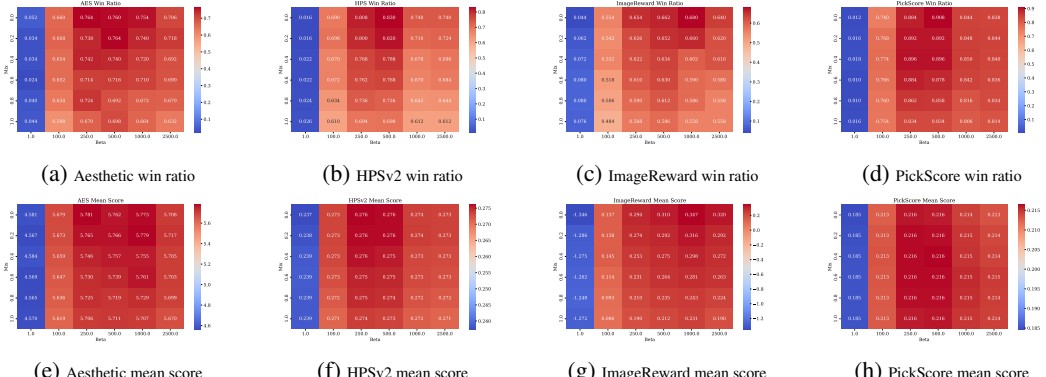

Figure 9: Visual example ablation study on hyper-parameter choice.

(a) Aesthetic win ratio (b) HPSv2 win ratio (c) ImageReward win ratio (d) PickScore win ratio

(e) Aesthetic mean score (f) HPSv2 mean score (g) ImageReward mean score (h) PickScore mean score

Figure 10: Heat map-based ablation study on hyper-parameter choice.

**Plug-and-play.** Our method is not only applicable to the original stable diffusion-based models and their fine-tuned versions optimized through preference optimization but also directly extends to high-quality stylized models fine-tuned on proprietary data. To demonstrate the versatility of our approach, we use the validation_unique dataset as our test benchmark prompts. As shown in Table 1, we observe significant improvements across various metrics. By fine-tuning the inference parameters, we enhance the performance of the DreamShaper model with 0.9B parameters, enabling it to surpass the best-performing methods on the 3B SDXL model in terms of aesthetic scores. Fig. 4 presents several comparative results, with notable improvements in structural integrity, contrast, and texture details.

## 5 CONCLUSIONS

In this paper, we investigate that previous preference optimization methods for diffusion models have overlooked the crucial role of unconditional/negative-conditional outputs in classifier-free guidance. We innovatively propose the task of Negative Preference Optimization as a plug-and-play inference enhancement strategy to achieve better preference-aligned generation. We summarize existing preference optimization training strategies and provide a straightforward but effective adaptation for Negative Preference Optimization. Extensive experimental results validate the effectiveness of Negative Preference Optimization.

**Limitations:** Diffusion-NPO requires the storage and loading of two different weight offsets for inference, which results in a higher storage cost. However, fortunately, preference optimization can typically be trained with LoRA, which requires only a minimal amount of additional storage.

ACKNOWLEDGEMENTS

This project is funded in part by National Key R&D Program of China Project 2022ZD0161100, by the Centre for Perceptual and Interactive Intelligence (CPII) Ltd under the Innovation and Technology Commission (ITC)'s InnoHK, by NSFC-RGC Project N_CUHK498/24. Hongsheng Li is a PI of CPII under the InnoHK.

REPRODUCIBILITY STATEMENT

We have undertaken substantial efforts to ensure that the results in this paper are reproducible. The training and evaluation code, along with detailed guidance, is made available at `https://github.com/G-U-N/Diffusion-NPO`. We believe these resources will aid in replicating our findings and foster further research that builds upon our contributions.

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

# APPENDIX

## I    RELATED WORKS

In this section, we give a brief introduction to previous efforts for diffusion-based preference optimization.

**Preference datasets and reward models.**  Previous works including Pick-a-pic (Kirstain et al., 2023), ImageReward (Xu et al., 2024), HPSv2 (Wu et al., 2023) collect image pairs generated by diffusion models with the same prompts and label the human preference for each pair. Laion-Aesthetic (Schuhmann, 2022) asks people to rate their preference for real images from 1 to 10. They then train the preference score models based on the preference label collected. These works lay a solid foundation for future human preference optimization works in diffusion models.

**Differentiable reward.** Some works including DRaFT (Clark et al., 2023), AlignProp (Prabhudesai et al., 2023), and ReFL (Xu et al., 2024) directly feed the generated images into pre-trained ImageReward models and update the generative model through the gradient of differentiable reward model. These works are straightforward and effective. However, due to the imperfection of reward models, these methods typically have reward leakage. For example, they may generate over-saturated images to cheat higher scores.

**Reinforcement learning.** Some works including DDPO (Black et al., 2023), and DPOK (Fan et al., 2024) propose to perceive the diffusion denoising process as a Markov decision process and apply the reinforcement learning algorithms for preference alignment. Some works (Zhang et al., 2024b) scale up the training for better performance. Generally, they apply PPO or the variants for training.

**Direct preference optimization.** Diffusion-DPO (Wallace et al., 2024) proposes a simulation-free training objective that enables direct preference optimization on preference-labeled image pairs. D3PO (Yang et al., 2024) and SPO (Liang et al., 2024b) combine reinforcement learning and direct preference optimization without the requirement to know specific score values for training.

**Other works related to negative preference optimization.** Beyond preference optimization in the realm of diffusion models, we've observed certain unpublished studies and efforts related to language models that have previously referenced or explored concepts akin to negative preference. We will present them in this discussion. Some community works  (Woolf, 2023; Euge, 2024; Nerfgun3, 2023) directly compress the negative preference representation into the negative-conditional inputs via textual inversion (Gal et al., 2022) to form a negative text embedding However, the negative preference is relatively complex, passively suppressing certain keyword-related features, exhibiting limited capability.  And language model-related works (Zhang et al., 2024a) on negative preference optimization aims to unlearn target data, usually bad concepts or privacy information, which optimizes a negative forget loss.

## II DISCUSSION

### II.1 DIFFUSION NPO IS FUNDAMENTALLY DIFFERENT FROM SCALING DPO

**Empirical validation.** The default training iteration for both DPO and NPO is defined as $T_0 = 2000$ iterations. To demonstrate the effectiveness of NPO compared to solely scaling up the existing DPO method, we adopt the official DPO code and extend its training iterations to $k \times T_0$, where $k$ ranges from 1 to 10 (i.e., up to $10 \times T_0 = 20,000$ iterations, equivalent to 10 times the training cost). In the following table, we denote the training iterations of each model as $k = 1, 2, \ldots, 10$. We summarize the quantitative evaluation results in Table 4. Our findings can be summarized as follows:

- Regardless of how long DPO is trained, the NPO weight offset, trained with only $1 \times T_0$ iterations, consistently and significantly improves overall performance.
- For example, DPO ($k = 1$) + NPO ($k = 1$) achieves an Aesthetic Score of 5.76, HPS Score of 27.46, and PickScore of 21.47, significantly outperforming DPO ($k = 10$), which requires a training cost of $10 \times T_0$—equivalent to $10/(1 + 1) = 5$ times longer than the combined cost of DPO ($k = 1$) + NPO ($k = 1$)—with scores of 5.626, 27.10, and 21.05, respectively.

We believe this provides strong evidence of the effectiveness of diffusion-NPO.

Table 4: Quantitative performance comparison of DPO and DPO + NPO across multiple metrics. All metrics are evaluated with official weights, where training iterations are denoted as $k \times T_0$ ($T_0 = 2000$). Avg denotes the average score, and Win denotes the average winning ratio against other methods.

| Method | Aesthetic | | HPS | | ImageReward | | PickScore | |
|---|---|---|---|---|---|---|---|---|
| | Avg | Win | Avg | Win | Avg | Win | Avg | Win |
| DPO ($k = 1$) | 5.6320 | 32.2% | 27.17 | 34.0% | 0.2576 | 42.6% | 21.00 | 18.6% |
| DPO ($k = 1$) + NPO ($k = 1$) | **5.7667** | **67.8%** | 27.46 | 66.0% | 0.3090 | 57.4% | 21.47 | **81.4%** |
| DPO ($k = 2$) | 5.6178 | 30.8% | 27.21 | 32.2% | 0.2674 | 41.8% | 21.06 | 19.6% |
| DPO ($k = 2$) + NPO ($k = 1$) | 5.7662 | **69.2%** | 27.51 | 67.8% | 0.3239 | 58.2% | 21.50 | 80.4% |
| DPO ($k = 3$) | 5.6214 | 27.8% | 27.28 | 33.4% | 0.3322 | 49.2% | 21.08 | 20.6% |
| DPO ($k = 3$) + NPO ($k = 1$) | 5.7685 | **72.2%** | 27.60 | 66.6% | 0.3362 | 50.8% | 21.54 | 79.4% |
| DPO ($k = 4$) | 5.6399 | 31.8% | 27.36 | 33.2% | 0.3574 | 46.0% | 21.12 | 21.6% |
| DPO ($k = 4$) + NPO ($k = 1$) | 5.7767 | 68.2% | 27.60 | 66.8% | 0.3612 | 54.0% | 21.55 | 78.4% |
| DPO ($k = 5$) | 5.6718 | 33.4% | 27.35 | 31.2% | 0.3540 | 44.4% | 21.15 | 18.6% |
| DPO ($k = 5$) + NPO ($k = 1$) | **5.7868** | 66.6% | **27.65** | **68.8%** | 0.3688 | 55.6% | **21.59** | **81.4%** |
| DPO ($k = 6$) | 5.6632 | 34.6% | 27.35 | 34.6% | 0.3664 | 45.8% | 21.18 | 20.6% |
| DPO ($k = 6$) + NPO ($k = 1$) | 5.7685 | 65.4% | 27.63 | 65.4% | 0.3780 | 54.2% | **21.59** | 79.4% |
| DPO ($k = 7$) | 5.6648 | 36.4% | 27.37 | 33.0% | 0.3980 | 45.0% | 21.19 | 21.0% |
| DPO ($k = 7$) + NPO ($k = 1$) | 5.7576 | 63.6% | **27.65** | 67.0% | 0.4141 | 55.0% | **21.59** | 79.0% |
| DPO ($k = 8$) | 5.6605 | 37.4% | 27.32 | 31.4% | 0.3840 | 49.8% | 21.18 | 19.4% |
| DPO ($k = 8$) + NPO ($k = 1$) | 5.7544 | 62.6% | 27.62 | **68.6%** | **0.4122** | 50.2% | 21.58 | 80.6% |
| DPO ($k = 9$) | 5.6438 | 37.6% | 27.22 | 27.2% | 0.3679 | 44.8% | 21.10 | 20.8% |
| DPO ($k = 9$) + NPO ($k = 1$) | 5.7463 | 62.4% | 27.61 | **72.8%** | 0.4121 | 55.2% | 21.56 | 79.2% |
| DPO ($k = 10$) | 5.6264 | 39.6% | 27.10 | 28.8% | 0.3214 | 42.0% | 21.05 | 19.6% |
| DPO ($k = 10$) + NPO ($k = 1$) | 5.7284 | 60.4% | 27.51 | 71.2% | 0.3986 | **58.0%** | 21.51 | 80.4% |

**Theoretical analysis.** Please note that in our motivation example, the expression $((1 - \gamma)(-\boldsymbol{\eta}))$ serves only as an approximation of the negative preference-optimized weight offset $\boldsymbol{\delta}$. Consequently, simply scaling $\boldsymbol{\eta}$ cannot be expected to fully replicate the effect of NPO. In the paper, we demonstrate that the weight adopted for unconditional or negative-conditional prediction can be expressed as

$$\boldsymbol{\theta}_{NPO} = \boldsymbol{\theta} + \alpha\boldsymbol{\eta} + \beta\boldsymbol{\delta},$$

where $\boldsymbol{\theta}$ represents the pretrained model weight, $\boldsymbol{\eta}$ is the weight offset derived from preference optimization, and $\boldsymbol{\delta}$ is the weight offset from negative preference optimization, with $\alpha$ and $\beta$ as their respective scaling factors. In our motivating example, we substitute $\boldsymbol{\delta}$ with $(1 - \alpha)(-\boldsymbol{\eta})$ to provide a simplified illustration of the core concept of negative preference optimization. However, this substitution assumes a parallel relationship between $\boldsymbol{\eta}$ and $\boldsymbol{\delta}$, which is an oversimplification.

In a more general framework, we can decompose $\boldsymbol{\delta}$ into two components:

$$\boldsymbol{\delta} = \boldsymbol{\delta}_{\parallel} + \boldsymbol{\delta}_{\perp},$$

Table 5: Comparison of NPO with SEG and Autoguidance on Stable Diffusion XL.

| Method | Aesthetic Avg | Win | HPSv2 Avg | Win | ImageReward Avg | Win | PickScore Avg | Win |
|---|---|---|---|---|---|---|---|---|
| SDXL | 6.11 | – | 27.89 | – | 0.62 | – | 22.06 | – |
| SEG | 6.15 | 55.2% | 26.26 | 10.6% | -0.10 | 19.4% | 20.99 | 9.0% |
| Autoguidance | 5.80 | 28.8% | 27.21 | 27.0% | 0.45 | 40.0% | 21.44 | 22.0% |
| NPO (Ours) | 6.11 | 51.4% | 28.78 | 81.2% | 0.92 | 73.6% | 22.69 | 82.0% |

where $\delta_\parallel$ is the component parallel to $\eta$, and $\delta_\perp$ is the orthogonal component. The orthogonal component $\delta_\perp$ cannot be captured by the simplified approximation. Specifically, according to the projection theorem, the parallel component can be computed as $\delta_\parallel = \mathrm{proj}_\eta(\delta) = \frac{\eta \cdot \delta}{\|\eta\|^2}\eta$, leaving $\delta_\perp = \delta - \delta_\parallel$. In the motivating example, we effectively set $\delta = (1 - \alpha)(-\eta)$, which is a scalar multiple of $-\eta$. Since scalar multiplication does not alter the direction of vectors, this implies $\delta_\perp = 0$ in the simplified case. However, in the general representation, $\delta_\perp$ exists and cannot be approximated or obtained through this scalar adjustment of $\eta$, highlighting the limitations of the simplified model and the nuanced contribution of NPO.

## II.2 EFFECTIVENESS OF NPO COMPARED TO TRAINING-FREE CFG-STRENGTHENING TECHNIQUES

To more comprehensively show the effectiveness of Diffusion-NPO, we compared Diffusion-NPO with training-free CFG-strengthening methods like Autoguidance (Karras et al., 2024) and SEG (Hong, 2025).

**Stable Diffusion XL.** We designed the following comparative tests: 1) SEG on Stable Diffusion XL vs. Naive CFG on Stable Diffusion XL. 2) DPO-optimized Stable Diffusion XL as conditional predictors and original Stable Diffusion XL as unconditional predictors vs. Naive CFG on Stable Diffusion XL. 3) DPO-optimized Stable Diffusion XL as conditional predictors and NPO-optimized Stable Diffusion XL as unconditional predictors vs. Naive CFG on Stable Diffusion XL. The evaluation results are shown in Table 5. SEG slightly improves Laion Aesthetic over SDXL but performs worse in other metrics, while Autoguidance produces blurry outputs due to the prediction gap between original and DPO-optimized SDXL, with NPO outperforming both across all metrics.

**Stable Diffusion v1-5.** We designed the following comparative tests: 1) DPO-optimized Stable Diffusion v1-5 as conditional predictors and original Stable Diffusion v1-5 as unconditional predictors vs. Naive CFG on DPO-optimized Stable Diffusion v1-5. 2) DPO-optimized Stable Diffusion v1-5 as conditional predictors and NPO-optimized Stable Diffusion v1-5 as unconditional predictors vs. Naive CFG on DPO-optimized Stable Diffusion v1-5. The results are shown in Table 6. NPO con-

Table 6: Comparison of NPO with Autoguidance on Stable Diffusion v1-5.

| Method | Aesthetic Avg | Win | HPSv2 Avg | Win | ImageReward Avg | Win | PickScore Avg | Win |
|---|---|---|---|---|---|---|---|---|
| DPO | 5.65 | – | 27.19 | – | 0.27 | – | 21.12 | – |
| Autoguidance | 5.63 | 51.2% | 26.96 | 34.6% | 0.13 | 42.0% | 21.26 | 58.2% |
| NPO | 5.76 | 68.8% | 27.60 | 76.6% | 0.31 | 59.2% | 21.58 | 84.4% |

sistently outperforms Autoguidance on Stable Diffusion v1-5 across all metrics, while SEG was not tested due to the lack of an open-source implementation.

## II.3 IMPACT OF TRAINING DATA ON NEGATIVE PREFERENCE PPTIMIZATION

In our original implementation of Diffusion-NPO, we maintained the same training data pairs, pre-processing strategies, and configurations as the original preference optimization (PO) techniques. This choice ensured a controlled comparison, allowing us to isolate the impact of the NPO method itself. However, a key question arises: can a more carefully curated negative dataset further enhance NPO's performance?

To investigate this, we conducted additional experiments focusing on the impact of training data quality in Diffusion-DPO-based NPO. Specifically, we explored two alternative approaches for generating negative preference data: 1) Perturbed Text Prompts: We randomly shuffled text prompt sequences within batches to create mismatched image-text pairs. This approach tests whether a

Table 7: Performance comparison of NPO trained with different data corruption strategies. Win-Rate represents the percentage of cases where the method outperforms the original NPO.

| Method | Aesthetic | | HPS | | ImageReward | | PickScore | |
|---|---|---|---|---|---|---|---|---|
| | Avg | Win | Avg | Win | Avg | Win | Avg | Win |
| Original NPO | 5.7621 | - | 27.60 | - | 0.3102 | - | 21.58 | - |
| NPO with Text Perturbation | 5.7407 | 45% | 27.52 | 37.6% | 0.3148 | 49.2% | 21.55 | 40.4% |
| NPO with Data Corruption | **5.7676** | **54%** | **27.63** | **51%** | **0.3222** | **54.0%** | **21.60** | **54.4%** |

more diverse but less semantically aligned negative dataset influences performance. 2) Random Image Corruptions: We applied controlled degradations, such as Gaussian blur, to negative images to analyze whether introducing noise helps refine NPO's learning process.

Our experimental results, summarized in Table 7, indicate that while text perturbations led to performance degradation, mild image corruptions slightly improved results. This suggests that while NPO benefits from a certain degree of data diversity, maintaining proper image-text alignment remains crucial. Notably, extreme corruptions resulted in significant output deviations, highlighting the trade-off between introducing variation and preserving meaningful training signals.

These findings reinforce the importance of data selection in negative preference training. While NPO can operate effectively using standard preference optimization datasets, strategic modifications to negative data can enhance its performance. This insight opens up new avenues for refining NPO through optimized data augmentation and curation strategies.

## II.4 ETHICAL GUIDELINES AND LIMITATIONS FOR RESPONSIBLE DIFFUSION-NPO APPLICATION

Diffusion-NPO enhances model outputs by steering generative models away from undesirable results, aligning with responsible AI principles. To ensure ethical deployment, the following guidelines must be observed:

- **Preventing Malicious Use**: Diffusion-NPO must be restricted to improving user experience and reducing harm, with safeguards against generating misinformation, bias, or deceptive content.
- **Transparent Disclosure**: Its implementation should be clearly documented to foster trust and accountability, especially in sensitive domains.
- **Ethical Oversight**: Regular audits and ethical reviews should assess potential unintended consequences to ensure responsible use.
- **Focused Scope**: Diffusion-NPO should be applied selectively in high-stakes areas to prevent overfitting and maintain broad applicability.

By adhering to these principles, Diffusion-NPO can effectively align generative models with ethical AI standards while minimizing risks.

## III MORE RESULTS

## Stable Diffusion

Prompt: "A beautiful 25 yearol old whos mother is from hong kong and father from turkey"

Prompt: "a woman in a silver suit with a ponytail, a detailed painting by WLOP, trending on Artstation, fantasy art, detailed painting, artstation hd, high detail"

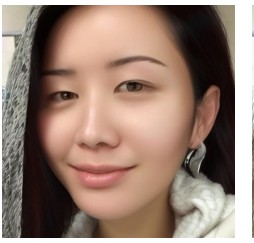 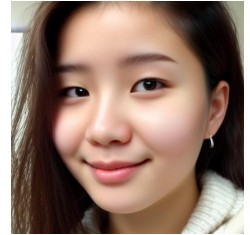 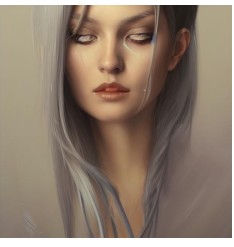 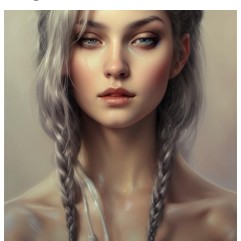

Prompt: "A house in the style of Escher"

Prompt: "Watercolour painting of an orange cat"

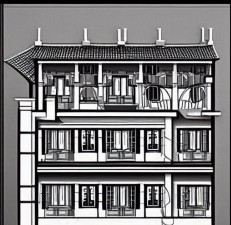 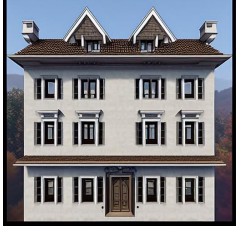

Prompt: "Milim, pink hair, that awesome time i got reincarnated as a slime"

Prompt: "Hyperrealistic full length portrait of gorgeous goddess l standing in field full of flowers … (over 30 words)"

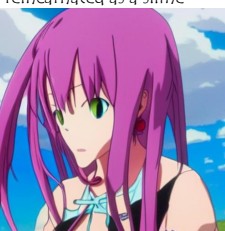 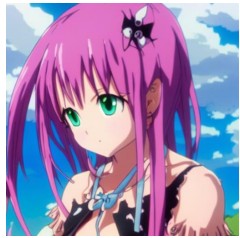 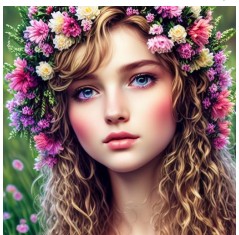

Prompt: "female face, blue jet green eyes, long hair, slant eyes, cheeky cheeks, smiling, carefree, … (over 20 words)"

Prompt: "A giant eagle monster art"

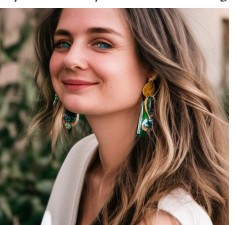 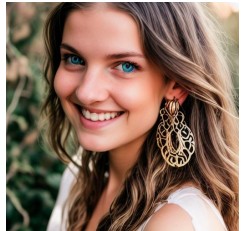 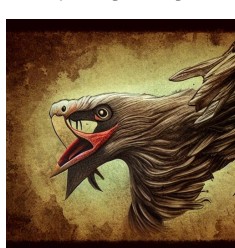 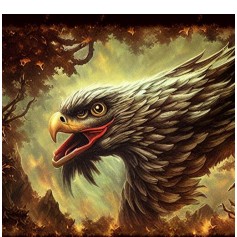

Prompt: "An anime woman"

Prompt: "Detailed painting of Attractive young women painting, model, detailed , cinematic lightning. "

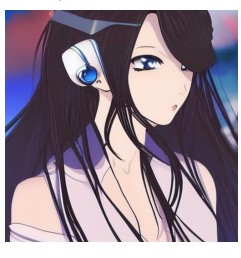 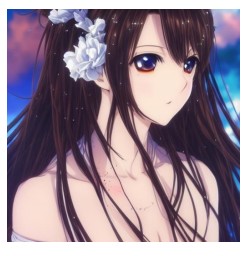 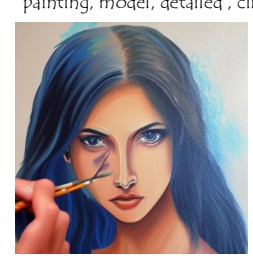 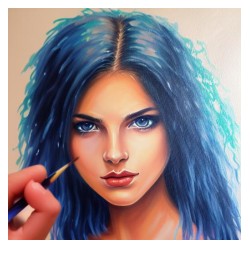

w/o NPO      w/ NPO      w/o NPO      w/ NPO

Figure 11: Comparison on Stable Diffusion 1.5.

## DreamShaper

Prompt: "A mermaid playing chess with a dolphin"

Prompt: "A living dragon on the world trade center"

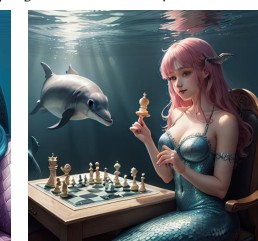
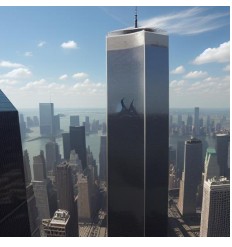
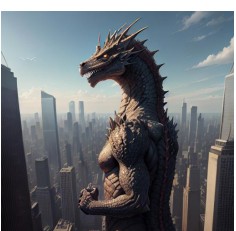

Prompt: "Dragon with six pairs of wings aterrorizing humans in a village near to a volcano"

Prompt:" Fast car"

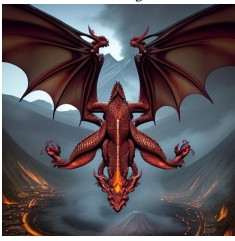
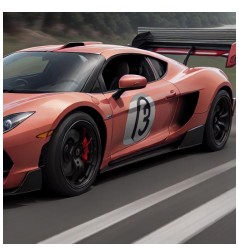

Prompt: "Painting of a black 22 year old girl with long braids, she has her eyes opened, highly detailed, style"

Prompt: "A panda riding a motorcycle"

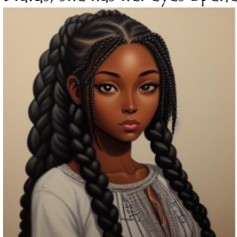
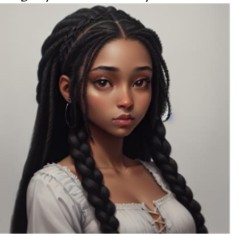
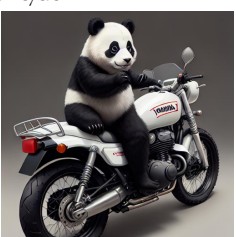

Prompt: "A dog and Santa Claus. Christmas trees in background.Black and white background"

Prompt: "An evil villain holding a mini Earth"

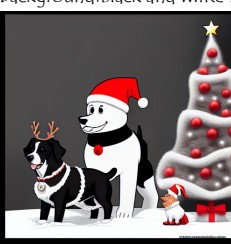
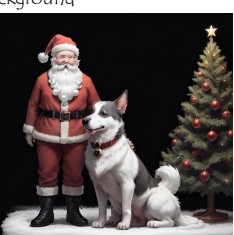
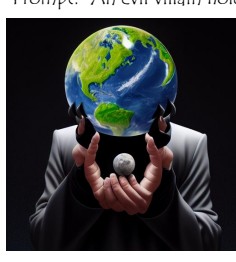
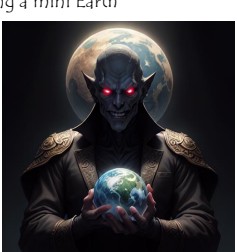

Prompt: "A jade statue of an adorable cat"

Prompt: "An anthropomorphic animal"

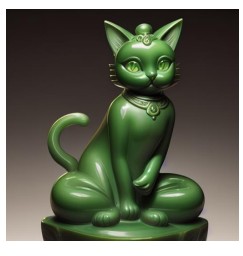
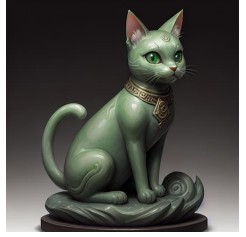
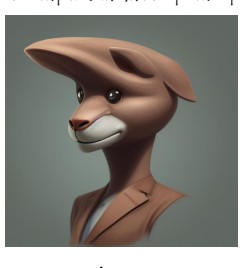
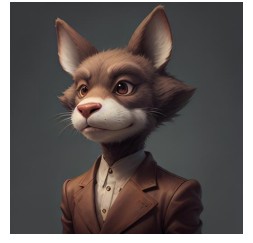

w/o NPO      w/ NPO      w/o NPO      w/ NPO

Figure 12: Comparison on DreamShaper.

## Stable Diffsuion XL

Prompt: "Human palm"

Prompt: "A cat, fat, chubby, very fine wispy and extremely long swirly wavy fur … (over 30 words)"

Prompt: "Jessica alba, anime style"

Prompt: "LeBron James slam dunking the planet saturn through its own rings"

Prompt: "A woman with blue eyes"

Prompt: "A gijinka black cat sushi chef"

Prompt: "A boss screaming at his employee for not working on the weekend by vincent van gogh"

Prompt: "Concept art, Disney, really crazy creature, colored pencils, cute, very creative drawing, … (over 30 words)"

Prompt: "A 20 yo girl in cyberpunk outfit"

Prompt: "Realistic photo with a light pink background color in various shades, a middle-aged … (over 30 words)"

w/o NPO          w/ NPO          w/o NPO          w/ NPO

Figure 13: Comparison on Stable Diffusion XL.

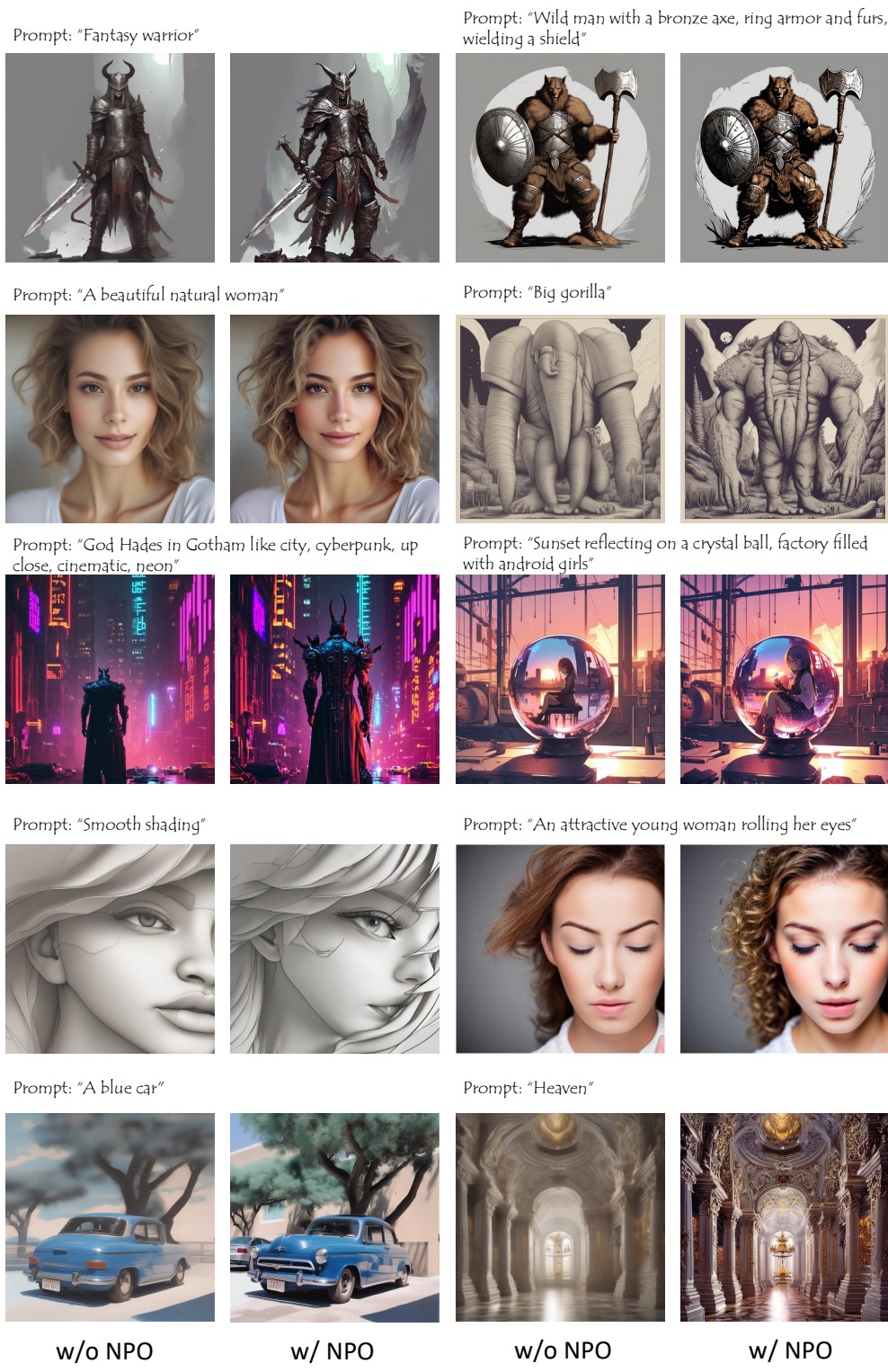

Figure 14: Comparison on Diffusion-DPO.

## Diffusion-SPO

Prompt: "A boss screaming at his employee for not working on the weekend by vincent van gogh"

Prompt: "Rachel Amber:1.5 wearing a black skirt. Thin body type, Young face, Sony Alpha A7 III, … (over 30 words)"

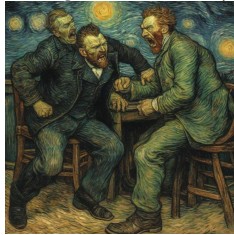 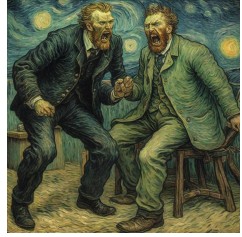 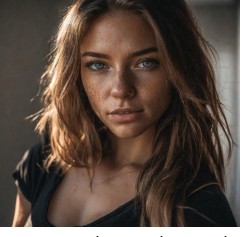 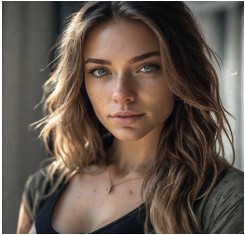

Prompt: "Japanese children ballet school"

Prompt: "Photorealistic style, photorealistic pope francis wearing drip footwear, drip tenis"

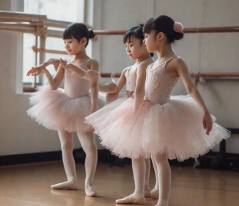 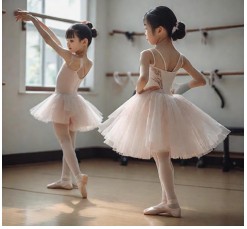 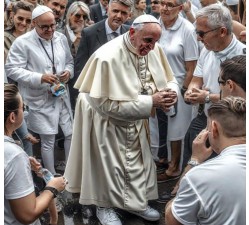

Prompt: "Michael jordan against bruce lee The straight blast round kick in the air nba basketball ball … (over 30 words)"

Prompt: "Random girl hugs Henry Cavill superman"

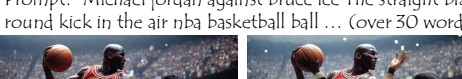

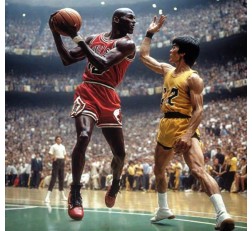 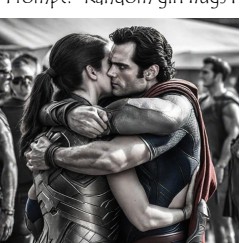 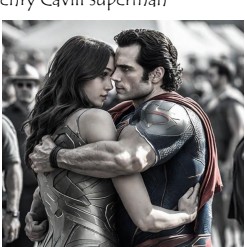

Prompt: "Highly detailed realistic photograph of a hand"

Prompt: "Sturdy and pink pickup truck"

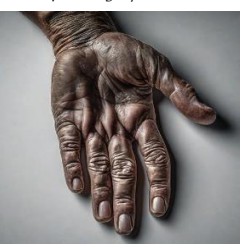 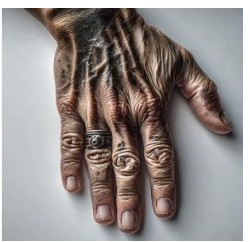 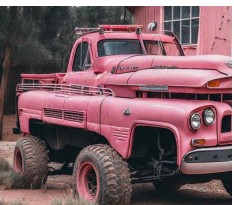 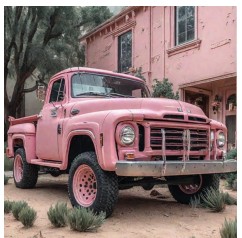

Prompt: "3d render of an ultrarealistic creature design, ONI entity with white long flowing hair"

Prompt: "A hot female Alex from Minecraft"

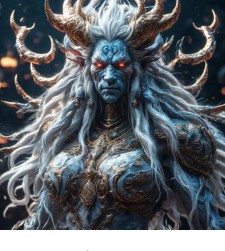 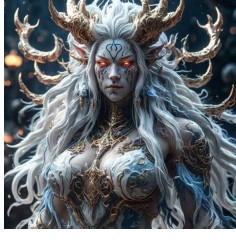 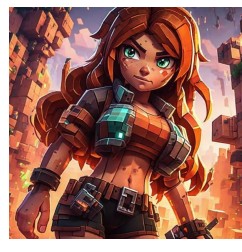 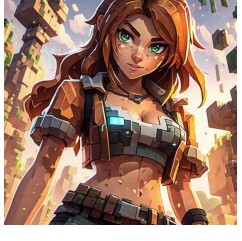

| w/o NPO | w/ NPO | w/o NPO | w/ NPO |

Figure 15: Comparison on Diffusion-SPO.

