# OpenReview forum: "Diffusion-NPO: Negative Preference Optimization for Better Preference Aligned Generation of Diffusion Models"
_ICLR.cc/2025/Conference — ICLR 2025 Poster_

### Official Review · Reviewer_5t3C · 2024-11-02

**Soundness:** 3
**Presentation:** 3
**Contribution:** 3
**Rating:** 6
**Confidence:** 3

**Summary:**

Many recent methods, such as diffusion DPO, fine-tune pretrained models using preference optimization. This paper highlights that while these methods focus on selecting preferred samples, they overlook the unconditional or negative outputs used in CFG. The paper addresses this by training a model using reverse DPO data for negative preference optimization, then incorporating this model into CFG to move outputs away from the negative values, resulting in improved image quality.

**Strengths:**

- The paper shows that using a model optimized with a preference for CFG's unconditional/negative term enhances performance.
- Qualitatively, the images generated with NPO show fewer artifacts and are sharper.
- This approach is simple to implement, as it only requires using the existing diffusion DPO method without additional data.

**Weaknesses:**

- The paper seems underdeveloped in certain areas. For instance, Table 2 displays scores but lacks discussion in the main text, and Figure 8’s caption does not specify the methods compared.
- It does not clearly demonstrate the difference from scaling up the existing DPO method. The results might be similar to running DPO for a longer period. In the motivating example of Section 2, δ appears proportional to η. Subtracting the resulting model's output seems equivalent to scaling η in the DPO model’s unconditional parameter.

**Questions:**

- Could you compare the results of this method to those of more intensive DPO training (e.g., running DPO for a longer period)?
- If the original pretrained model is used instead of the NPO-trained model to obtain unconditional/negative outputs, is there a significant performance drop?

---

> ### Author Response · Authors · 2024-11-23
> **Response to Reviewer 5t3C [1/3]**
>
> Thank you for your in-depth review and question. We value the opportunity to further showcase the effectiveness of Diffusion-NPO.
>
>
>
> > **W1** "The paper seems underdeveloped in certain areas." "Table 2 displays scores but lacks discussion in the main text."
>
> Please refer to lines 411-414 where we provided a discussion of Table 2. Following your suggestion, we are willing to enrich the discussion here.
>
> "Figure 8’s caption does not specify the methods compared."
>
> Figure 8 is to show user study comparison of samples generated with NPO and samples generated without NPO. We adopted the legends with different colors to show the difference. The legends are "w/ NPO better" (samples generated with NPO are better), "w/o NPO better" (samples generated without NPO are better), and "Tie" (No preference).
>
> Additionally, please refer to lines 470-473, where we provided a discussion on Figure 8 and specified the models adopted including Stable Diffusion XL, Stable Diffusion v1-5, and Dreamshaper.
>
> > **W2.1** "In the motivating example of Section 2. Subtracting the resulting model's output seems equivalent to scaling η in the DPO model’s unconditional parameter."
>
> Please kindly note that the $((1 − \gamma)(− \boldsymbol \eta))$ is just an approximate of the negative preference optimized $\boldsymbol \delta$, therefore we can not expect simpling scaling $\eta$ to achieve the same effect of NPO.
>
> In the paper, we showed that the weight adopted for unconditional/negative-conditional prediction can be denoted as
>
> $$
> \boldsymbol \theta_{NPO} = \boldsymbol \theta + \alpha\boldsymbol \eta + \beta\boldsymbol \delta,
> $$
>
> where $\boldsymbol \theta$ is the pretrained model weight, $\boldsymbol \eta$ is the weight offset obtained through preference optimization, and $\boldsymbol \delta$ is the weight offset obtained through negative preference optimization.  In the motivation example, we replace $\boldsymbol \delta $ with $(1-\alpha) (-\boldsymbol \eta)$ to give a straightforward thinking and illustration of core idea of negative preference optimization. However, please note that it is just a simplified case that assumes the parallel of $\boldsymbol \eta$ and $\boldsymbol \delta$.
>
>  In general, we can decompose the components of $\boldsymbol \delta$ into
> $$
> \boldsymbol \delta = \boldsymbol \delta_{\parallel} + \boldsymbol \delta_{\perp}.
> $$
> **The component $\boldsymbol \delta_{\perp}$ can never be obtained with the oversimplified case.**

---

> ### Author Response · Authors · 2024-11-23
> **Response to Reviewer 5t3C [2/3]**
>
> > **W2.2** "It does not clearly demonstrate the difference from scaling up the existing DPO method. "
>
>
>
> The default training iteration of DPO and NPO are both 2,000 iterations. To show the effectiveness of NPO compared to solely scaling up existing DPO method. We adopt the official code of DPO and extend the training iteration of DPO to 20,000 iterations (i.e., 10x training cost). In the following table, we use the 1x, 2x, ... to denote the training iterations of each model.
>
> We summarize the quantitative evaluation results in the following table.
>
> | Method               | AES Mean | AES Win Rate | HPS Mean | HPS Win Rate | ImageReward Mean | ImageReward Win Rate | PickScore Mean | PickScore Win Rate |
> | -------------------- | -------- | ------------ | -------- | ------------ | ---------------- | -------------------- | -------------- | ------------------ |
> | DPO (1x)             | 5.6320   | 32.2%        | 27.17    | 34.0%        | 0.2576           | 42.6%                | 21.00          | 18.6%              |
> | DPO (1x) + NPO (1x)  | 5.7667   | 67.8%        | 27.46    | 66.0%        | 0.3090           | 57.4%                | 21.47          | 81.4%              |
> | DPO (2x)             | 5.6178   | 30.8%        | 27.21    | 32.2%        | 0.2674           | 41.8%                | 21.06          | 19.6%              |
> | DPO (2x) + NPO (1x)  | 5.7662   | 69.2%        | 27.51    | 67.8%        | 0.3239           | 58.2%                | 21.50          | 80.4%              |
> | DPO (3x)             | 5.6214   | 27.8%        | 27.28    | 33.4%        | 0.3322           | 49.2%                | 21.08          | 20.6%              |
> | DPO (3x) + NPO (1x)  | 5.7685   | 72.2%        | 27.60    | 66.6%        | 0.3362           | 50.8%                | 21.54          | 79.4%              |
> | DPO (4x)             | 5.6399   | 31.8%        | 27.36    | 33.2%        | 0.3574           | 46.0%                | 21.12          | 21.6%              |
> | DPO (4x) + NPO (1x)  | 5.7767   | 68.2%        | 27.60    | 66.8%        | 0.3612           | 54.0%                | 21.55          | 78.4%              |
> | DPO (5x)             | 5.6718   | 33.4%        | 27.35    | 31.2%        | 0.3540           | 44.4%                | 21.15          | 18.6%              |
> | DPO (5x) + NPO (1x)  | 5.7868   | 66.6%        | 27.65    | 68.8%        | 0.3688           | 55.6%                | 21.59          | 81.4%              |
> | DPO (6x)             | 5.6632   | 34.6%        | 27.35    | 34.6%        | 0.3664           | 45.8%                | 21.18          | 20.6%              |
> | DPO (6x) + NPO (1x)  | 5.7685   | 65.4%        | 27.63    | 65.4%        | 0.3780           | 54.2%                | 21.59          | 79.4%              |
> | DPO (7x)             | 5.6648   | 36.4%        | 27.37    | 33.0%        | 0.3980           | 45.0%                | 21.19          | 21.0%              |
> | DPO (7x) + NPO (1x)  | 5.7576   | 63.6%        | 27.65    | 67.0%        | 0.4141           | 55.0%                | 21.59          | 79.0%              |
> | DPO (8x)             | 5.6605   | 37.4%        | 27.32    | 31.4%        | 0.3840           | 49.8%                | 21.18          | 19.4%              |
> | DPO (8x) + NPO (1x)  | 5.7544   | 62.6%        | 27.62    | 68.6%        | 0.4122           | 50.2%                | 21.58          | 80.6%              |
> | DPO (9x)             | 5.6438   | 37.6%        | 27.22    | 27.2%        | 0.3679           | 44.8%                | 21.10          | 20.8%              |
> | DPO (9x) + NPO (1x)  | 5.7463   | 62.4%        | 27.61    | 72.8%        | 0.4121           | 55.2%                | 21.56          | 79.2%              |
> | DPO (10x)            | 5.6264   | 39.6%        | 27.10    | 28.8%        | 0.3214           | 42.0%                | 21.05          | 19.6%              |
> | DPO (10x) + NPO (1x) | 5.7284   | 60.4%        | 27.51    | 71.2%        | 0.3986           | 58.0%                | 21.51          | 80.4%              |
>
> From the table, we can observe the following:
>
> 1. Regardless of how long DPO is trained, the NPO weight offset, trained with only 1x training cost, consistently enhances overall performance.
>
> 2. The combination of **DPO (1x) + NPO (1x)** achieves significantly better results:
>    - **Aesthetic Score**: 5.7667
>    - **HPS Score**: 27.46
>    - **PickScore**: 21.47
>
>    This outperforms **DPO (10x)**, which requires 5 times longer training cost (10/(1+1)) but only achieves:
>    - **Aesthetic Score**: 5.6264
>    - **HPS Score**: 27.10
>    - **PickScore**: 21.05
>
> **We believe this is very strong evidence of the effectiveness of diffusion-NPO.**

---

> > ### Comment · Reviewer_5t3C · 2024-11-25
> >
> > Thank you for providing a detailed rebuttal to my initial review. I also apologize for missing some of the explanations regarding certain tables and figures. Initially, I thought the method was too simple and almost indistinguishable from DPO. However, based on the experiments presented by the authors, it seems clear that the results differ significantly from DPO even when run for an extended time. Moreover, the fact that the method can be applied beyond DPO demonstrates a clear distinction.
> >
> > The main point of my initial review was the differentiation from DPO, and I believe this has been sufficiently addressed in the rebuttal. As a result, I will raise my score.
> >
> > By the way, could you provide a more detailed explanation regarding W2.1? I am a bit confused about what the notation means and what is implied by it being "parallel."

---

> > > ### Author Response · Authors · 2024-11-25
> > > **Sincerely thank you for raising the score!**
> > >
> > > We are grateful for your decision to raise the score. Your recognition and feedback mean a lot to us.
> > >
> > >
> > > For W2.1, we denote the $\boldsymbol \theta_{NPO}$ as $\boldsymbol \theta_{NPO} = \boldsymbol \theta + \alpha \boldsymbol \eta  + \beta  \boldsymbol \delta $, where $\boldsymbol \theta$ is the pretrained model weight, $\boldsymbol \eta$ is the weight offset obtained through preference optimization, $\boldsymbol \delta$ is the weight offset obtained through negative preference optimization, and $\alpha$ and $\beta$ are the scale factors for $\boldsymbol \eta$ and $\boldsymbol \delta$, respectively.
> > >
> > > Our motivating example can be viewed as a specific case of the above general representation. Specifically, in the motivating example, we have  $\boldsymbol \theta_{NPO} =  \boldsymbol \theta + \gamma \eta = \boldsymbol \theta + \boldsymbol \eta + (1-\gamma) (- \boldsymbol \eta) $.
> > >
> > > It is equivalent to
> > >
> > > - $\alpha=\beta =1$
> > > - $\boldsymbol \eta = \boldsymbol \eta$
> > > - $ \boldsymbol \delta = (1-\gamma) (-\boldsymbol \eta)$ $=$ [some scalar] $ \times \boldsymbol \eta$
> > >
> > > **Note that the scalar multiplication will not change the direction of vectors.** Therefore, we have $\boldsymbol \delta \parallel \boldsymbol \eta$ in the simplified case.
> > >
> > > Return back to our general representation, for any $\boldsymbol \eta$ and $\boldsymbol \delta$, we can decompose $\boldsymbol \delta$  into two orthogonal components. Specifically, we can compute the component of $\boldsymbol \delta$ parallel to $\boldsymbol \eta$, which can be denoted as $\boldsymbol \delta_{\parallel} = \frac{\boldsymbol{\delta} \cdot \boldsymbol{\eta}}{\|\boldsymbol{\eta}\|^2} \boldsymbol{\eta}$ according to the projection theorem. Then we have $\boldsymbol \delta_{\perp} = \boldsymbol \delta - \boldsymbol \delta_{\parallel}$.  Here, $\boldsymbol \delta_{\perp}$ can never be obtained or approximated through this simplified case.
> > >
> > >
> > > Again, we appreciate your decision to raise the score. We are always here should you have any further questions or need additional information.
> > >
> > > Best regards,
> > >
> > > The Authors

---

> ### Author Response · Authors · 2024-11-23
> **Response to Reviewer 5t3C [3/3]**
>
> > **Q1** "Could you compare the results of this method to those of more intensive DPO training (e.g., running DPO for a longer period)"
>
> Please refer to W2.2.
>
> > **Q2** "If the original pretrained model is used instead of the NPO-trained model to obtain unconditional/negative outputs, is there a significant performance drop?"
>
> Yes, it will suffer from significant performance degradation. Please refer to Q1 of Reviewer 5Mi5. Your mentioned baseline is equivalent to the baseline of AutoGuidance [1].
>
>
>
> **Further clarity on the application scope of Diffusion-NPO**:
>
>  In addition to the above replies to proposed questions, **we hope to clarify that our method and validation are not solely limited to DPO (Direct Preference Optimization) which requires paired data for tuning**. Please refer to lines 66-81 and lines 354-367. Specifically, current diffusion-based preference optimization methods can be generally categorized into three types: 1) Differentiable Reward (DR) 2) Reinforcement Learning (RL)  3) Direct Preference Optimization (DPO). We validate diffusion-NPO on three baselines: VADER (DR-based method, online generation, and evaluation, data-free), Diffusion-SPO (RL-based method, online generation, and evaluation, data-free), Diffusion-DPO (DPO-based method). This is to say, **we implement and validate the effectiveness of NPO on all three types of methods for diffusion-based preference optimization instead of being limited to DPO**.
>
>
>
> [1] Guiding a Diffusion Model with a Bad Version of Itself. NeurIPS Oral.

---

> > ### Comment · Area_Chair_a7xu · 2024-11-24
> > **Discussion Period Ending Soon**
> >
> > Dear Reviewer,
> >
> > The discussion period will end soon. Please take a look at the author's comments and begin a discussion.
> >
> > Thanks, Your AC

---

> ### Author Response · Authors · 2024-11-25
> **Discussion deadline is approaching**
>
> Dear Reviewer 5t3C,
>
> As the discussion deadline approaches, we kindly ask if our response has addressed your concerns. Please feel free to share any additional questions or feedback, and we’ll be happy to provide further clarification.
>
> Best regards,
>
> The Authors

---

### Official Review · Reviewer_5Mi5 · 2024-11-04

**Soundness:** 4
**Presentation:** 4
**Contribution:** 3
**Rating:** 8
**Confidence:** 4

**Summary:**

This paper tackles the task of aligning diffusion-based generative models with human preferences. The authors propose training an additional model that aligns with the opposite of human preferences. At inference, this model serves as the unconditional or negative-conditional component in classifier-free guidance. The method is simple to implement—requiring only a reward inversion (by multiplying it by -1) for reward-based methods or reversing the order of preferred image pairs for DPO-based methods. Evaluations on various text-to-image and text-to-video models show both qualitative and quantitative improvements across the board.

**Strengths:**

- The main idea is very intuitive, simple to implement, and highly effective.
- The method is compared against several baseline alignment methods on multiple image and video diffusion models. The proposed algorithm is shown to be an improvement using various quality metrics and human user studies.
- The proposed technique is quite general, and can be used alongside any alignment algorithm.

**Weaknesses:**

The paper doesn’t introduce a new alignment technique; instead, it builds on existing alignment algorithms to train the negative-aligned model. This has both pros and cons: on the plus side, it can work alongside any alignment method, but on the downside, its quality is limited by the performance of the alignment algorithm used.

**Questions:**

Recently, several works have improved text-to-image models by changing the unconditional or negative-conditional component in classifier-free guidance (CFG). The general idea is to use a “worse” version of the model for the second part of CFG. For example, Autoguidance[1] uses an earlier checkpoint in training, and SEG[2] manually corrupts some internal features in the model’s attention layers. Some of these methods don’t need extra training, so it would be interesting to compare them to Diffusion-NPO. While the paper already makes a similar comparison summarized in Figure 5, expanding on this could help show how much improvement comes from using negative preferences versus just a “worse” model.

[1] Karras, Tero, et al. "Guiding a Diffusion Model with a Bad Version of Itself." *arXiv preprint arXiv:2406.02507* (2024).

[2] Hong, Susung. "Smoothed Energy Guidance: Guiding Diffusion Models with Reduced Energy Curvature of Attention." *arXiv preprint arXiv:2408.00760* (2024).

---

> ### Author Response · Authors · 2024-11-23
> **Thank you for your thoughtful review [1/1]**
>
> Thank you for your encouraging review!
>
> > **Q1**
>
> Thank you for proposing the insightful question. We conduct additional experiments to show the effectiveness of NPO compared to previous proposed training-free CFG-strengthening techniques including Autoguidance [1] and SEG [2] as you mentioned.
>
> Specifically, we test with the following comparisons:
>
> 1. SEG on Stable Diffusion XL  vs Naive CFG on Stable Diffusion XL.
> 2. DPO-optimized Stable Diffusion XL as conditional predictors and original Stable Diffusion XL as the unconditional predictors vs Naive CFG on Stable Diffusion XL.
> 3. DPO-optimized Stable Diffusion XL as conditional predictors and NPO-optimized Stable Diffusion XL as unconditional predictors vs Naive CFG on Stable Diffusion XL.
>
> Note that the original Stable Diffusion XL can be viewed as an earlier version of DPO-optimized Stable Diffusion XL. Therefore, we adopt 2 as the baseline of Autoguidance.
>
> We generate all the images with the same seed and the same prompts from the test_unique split as the text prompts. We score generated images with the HPSv2, PickScore, ImageReward, and Laion Aesthetic and compare their average scores and win-rates.
>
> The results are shown in the following table. All the win-rate are compared with original SDXL.
>
> | Method       | AES Mean Score | AES Win-Rate | HPS Mean Score  | HPS Win-Rate | ImageReward Mean Score | ImageReward Win-Rate | PickScore Mean Score | PickScore Win-Rate |
> | ------------ | -------------- | ------------ | ------------------ | ------------ | ---------------------- | -------------------- | ------------------------ | ------------------ |
> | SDXL         | 6.1142         | -            | 27.89              | -            | 0.6246                 | -                    | 22.06                    | -                  |
> | SEG          | 6.1498         | 55.2%        | 26.26              | 10.6%        | -0.1042                | 19.4%                | 20.99                    | 9.0%               |
> | AutoGuidance | 5.8049         | 28.8%        | 27.21              | 27.0%        | 0.4543                 | 40.0%                | 21.44                    | 22.0%              |
> | NPO (Ours)   | 6.1116         | 51.4%        | 28.78              | 81.2%        | 0.9210                 | 73.6%                | 22.69                    | 82.0%              |
>
> We can see that SEG only achieves minor improvement on Laion-Aesthetic compared to the original baseline of SDXL. All the other metrics show that SEG significantly underperforms the naive CFG strategy with SDXL. This can be attributed to that SEG was originally proposed to tackle the circumstances where the prompt is not used. When applying AutoGuidance, we find the outputs tend to be blurry. Our quantitative evaluation also proves our claim. This can be attributed to the prediction difference between original SDXL and DPO-optimized SDXL being too large (as analyzed in lines 340-344)
>
>
>
> We also test on the Stable Diffusion v1-5 for further comparison:
>
> 1. DPO-optimized Stable Diffusion v1-5 as conditional predictors and original Stable Diffusion v1-5 as the unconditional predictors vs Naive CFG on DPO-optimized Stable Diffusion v1-5.
>
> 2. DPO-optimized Stable Diffusion v1-5 as conditional predictors and NPO-optimized Stable Diffusion v1-5 as the unconditional predictors vs Naive CFG on DPO-optimized Stable Diffusion v1-5.
>
> We did not test SEG on Stable Diffusion v1-5 since we did not find open-source implementation of SEG on Stable Diffusion v1-5.
>
> The results are shown in the following table. All the win-rate are compared with dpo-optimized Stable Diffusion v1-5.
>
> | Method       | AES Mean Score | AES Win Rate | HPS Mean Score | HPS Win Rate | ImageReward Mean Score | ImageReward Win Rate | PickScore Mean Score | PickScore Win Rate |
> | ------------ | -------------- | ------------ | ------------------ | ------------ | ---------------------- | -------------------- | ------------------------ | ------------------ |
> | DPO          | 5.6484         | -            | 27.19              | -            | 0.2652                 | -                    | 21.12                    | -                  |
> | AutoGuidance | 5.6277         | 51.2%        | 26.96              | 34.6%        | 0.1251                 | 42.0%                | 21.26                    | 58.2%              |
> | NPO          | 5.7621         | 68.8%        | 27.60              | 76.6%        | 0.3102                 | 59.2%                | 21.58                    | 84.4%              |
>
> NPO still consistently outperforms the compared baselines.

---

> > ### Comment · Reviewer_5Mi5 · 2024-11-26
> > **Thanks to the authors for the reply**
> >
> > I'd like to thank the authors for their detailed follow-up. It's very interesting to see how well the proposed method performs compared to learning-free guidance approaches, especially given its simplicity. All my questions have been fully addressed, and I will maintain my score and advocate for the acceptance of this paper.

---

### Official Review · Reviewer_Fyov · 2024-11-04

**Soundness:** 3
**Presentation:** 3
**Contribution:** 3
**Rating:** 6
**Confidence:** 4

**Summary:**

This paper introduces Negative Preference Optimization (NPO), a novel approach for enhancing the preference alignment of diffusion models by explicitly training the model to recognize and avoid generating outputs that are misaligned with human preferences.The proposed NPO aims to improve diffusion model outputs by creating a separate model component trained to avoid generating undesirable features.  By training models to understand both positive and negative preferences, NPO enhances the effectiveness of classifier-free guidance, which relies on balancing conditional and negative-conditional outputs to improve output quality.The writers designed the NPO to integrate easily with existing diffusion models, such as SD1.5, SDXL, and various preference-optimized versions, improving image and video quality across these models without requiring significant modifications.The paper provides quantitative and qualitative evaluations across several metrics (HPSv2, ImageReward, PickScore, Laion-Aesthetic) and demonstrates that NPO consistently enhances image quality, color accuracy, structural coherence, and alignment with human aesthetic preferences. Overall, this work addresses a gap in preference optimization by focusing on both desirable and undesirable outputs, improving human-aligned generation across diverse applications in image and video synthesis.

**Strengths:**

This paper introduces a novel technique called Negative Preference Optimization (NPO) to improve the alignment of diffusion models with human preferences. Unlike traditional methods that focus solely on desirable features, NPO addresses the problem of undesirable outputs by training the model to recognize and avoid them. This innovative approach is both creative and practical, as it leverages existing preference data by simply reversing image pair rankings. The paper demonstrates that this simple technique can significantly improve the effectiveness of classifier-free guidance without requiring complex new datasets or training procedures.

The paper is well-executed, providing both theoretical insights and practical validation. It offers a clear and well-organized presentation, with a logical flow from problem statement to methodology and experimental results. The proposed method is grounded in a thoughtful analysis of classifier-free guidance, recognizing the critical role of conditional and negative-conditional outputs in achieving preference alignment. This theoretical foundation is reinforced by a comprehensive suite of quantitative evaluations (using metrics like HPSv2, ImageReward, and PickScore) and qualitative comparisons (sample images with and without NPO) that effectively demonstrate the improvements NPO brings to various diffusion models, including SD1.5, SDXL, and video diffusion models. The experiments are carefully structured to show the plug-and-play nature of NPO, further adding to the paper’s quality. Although some technical concepts may be challenging, the paper is structured to guide readers through them systematically.

The significance of this work is high, as it provides an effective, adaptable solution to a widely recognized issue in diffusion-based image and video generation: producing outputs that align well with human aesthetics and avoid undesired qualities. NPO addresses a core limitation in preference optimization approaches—specifically, the lack of attention to undesirable outputs—while remaining compatible with popular models and training frameworks. The paper’s contributions are likely to influence future work in human preference alignment, particularly in fields where aesthetic quality and user satisfaction are critical, such as digital art, content creation, and interactive AI. Furthermore, NPO’s plug-and-play compatibility makes it a practical choice for both researchers and developers aiming to improve generation quality without extensive re-training or model modification, boosting its applicability across the field.

Overall, the paper presents an original, well-supported, and clearly articulated contribution that addresses a key gap in diffusion model preference optimization, with practical implications for a broad range of image and video generation applications.

**Weaknesses:**

1. While the paper's introduction of Negative Preference Optimization (NPO) is innovative, the simple reversal of preference pair rankings may oversimplify the complexity of human aesthetics. Negative preferences are not always straightforward opposites of positive preferences, and undesirable features can be subtle or context-dependent.

2. Metrics-driven approach lacking user perspective

3. The paper validates NPO primarily on general-purpose datasets and models like Stable Diffusion and DreamShaper. While these are commonly used in text-to-image generation, they may not represent the variety of domains where preference alignment is crucial, such as medical imaging, scientific visualization, or highly specialized art styles.

4. The performance implications of NPO's dual weight system, especially for large-scale or real-time applications, are not fully explored.

5. the paper's focus on CFG-based models limits the exploration of NPO's compatibility with other diffusion architectures.
NPO's compatibility with non-CFG diffusion models remains unclear.

**Questions:**

1. How does NPO distinguish between truly negative attributes and those that are simply neutral or contextual?
2. What are the computational costs of NPO, especially in terms of memory and processing time for real-time applications?
3. How sensitive is NPO's performance to parameter settings, and what are the recommended heuristics for tuning them effectively?
4. How well does NPO perform on domain-specific datasets like medical imaging or abstract art?
5. Beyond quantitative metrics, what is the perceived quality improvement from NPO based on qualitative user evaluations?
6. How does NPO mitigate the risk of reinforcing biases present in preference datasets?
7. Have you discussed ethical guidelines and limitations on NPO's use to ensure responsible application?

---

> ### Author Response · Authors · 2024-11-23
> **Response to Reviewer Fyov [1/4]**
>
> Thank you for your detailed comments! We are delighted that you find our paper novel, simple and effective, clear and well-organized, and with **future impact**!
>
> > **W1** "While the paper is innovative, the simple reversal of preference pair rankings may oversimplify the complexity of human aesthetics."
>
> Thank you for your thoughtful comment!
>
> On one hand, the reversal of preference rankings naturally selects the "less-preferred" option in a pair, a concept inherently implied when individuals choose their "preferred" option. This makes it a straightforward and reasonable approach to construct negative data for training.
>
> On the other hand, our approach is intentionally designed as a deliberate simplification to tackle a specific challenge: steering the model away from generating outputs that misalign with user preferences. By reversing the rankings, we effectively penalize undesired outputs, reducing their likelihood without significantly increasing computational complexity.
>
> **We view our method as a foundational step rather than a comprehensive solution.** It highlights the potential of leveraging preference data in innovative ways while acknowledging that more sophisticated methods could better capture the nuances of human aesthetics.
> Future research could explore incorporating richer datasets, utilizing advanced ranking models, or developing mechanisms to account for context-dependent aesthetic judgments. These directions could further enhance the model's ability to generate outputs that more accurately align with the complexity and subjectivity of human preferences.
>
> > **W2** "Metrics-driven approach lacking user perspective."
>
> We already conducted a comprehensive user study that evaluates key aspects such as high-frequency details, color and lighting, and low-level compositional structures in our original submission. Please refer to Lines 430-475 and Figure 8 in the paper.
>
> The user study highlights that NPO significantly enhances high-frequency details in the generated outputs, produces colors and lighting that better reflect human preferences, and improves the overall compositional structure of the images. These findings underscore the effectiveness of NPO in aligning diffusion models with human aesthetics.
>
> Additionally, the reward models used in our paper, HPS v2 and PickScore, are specifically designed to incorporate the user's perspective. By leveraging these models, our goal is to ensure the generated outputs align more closely with human preferences.
>
>
> > **W3**  "The paper validates NPO primarily on general-purpose datasets, While these are commonly used in text-to-image generation, they may not represent the variety of domains where preference alignment is crucial, such as medical imaging, scientific visualization, or highly specialized art styles."
>
>
> Mainstream prior works [1-8] on diffusion-based preference optimization predominantly focus on general-purpose text-to-image or text-to-video generation tasks, using these as standard benchmarks for validation.
>
> **Consistent with this established practice**, our study evaluates performance on these widely adopted tasks, validating our effectiveness on both text-to-image and text-to-video. Our primary aim is to improve human preference alignment within the context of general-purpose text-to-image and text-to-video generation following the common practice of prior works.
>
> While extending preference alignment to specialized domains such as medical imaging, scientific visualization, or highly specialized art styles may hold potential value, such explorations **might lie beyond the scope of our current work**. Specifically, to the best of our knowledge, we do not find relevant datasets, reward models, or methods for preference optimization in the field of medical imaging. The concept of preference optimization in the field of medical imaging is also somewhat ambiguous, as there are no established definitions or frameworks specifically addressing what constitutes "preference" in the context of medical imaging. This raises questions about how preference can be meaningfully defined and evaluated in such a domain.
>
>
> Additionally, as none of the compared works have conducted experiments in these small specific domains, **focusing on general-purpose benchmarks allows for a fair and meaningful comparison**.
>
> We acknowledge that future research could explore these specialized areas to further broaden the applicability of diffusion-based preference optimization.

---

> ### Author Response · Authors · 2024-11-23
> **Response to Reviewer Fyov [2/4]**
>
> > **W4**  "NPO for Large-scale and real-time applications are not fully explored."
>
> **Large-Sacle:**
>
> We are already using the large-scale models to validate our paper.
> In total, we validate our methods on **four large-scale models** including Stable Diffusion v1-5 (0.9 B), DreamShaper (0.9 B), Stable Diffusion XL (3 B), VideoCrafter v2 (1.5 B), which are all considerable large models in diffusion fields.  For example, the largest model in the widely recognized DiT (SoTA performance on the diffusion-based image generation) termed as DiT-XL/2 has only 0.6 B parameters.
>
> For a direct comparison, most previous works [4-8] on diffusion-based preference optimization only validate their methods on **one model** including Stable Diffusion v1-4, v1-5, or v2, which are all 0.9 B parameters.
>
>
>
> **Real-Time:**
> This paper focuses on improving model alignment with human preferences, rather than developing real-time applications.
>
> Diffusion models are inherently slow due to their iterative denoising process, making them unsuitable for real-time applications under standard settings. Additionally, none of the prior works [1-8] have achieved real-time performance. Therefore, real-time application is not explored in this paper.
>
>
>
> > **W5** "Focus on CFG-based models limits the exploration of NPO's compatibility with other diffusion architectures. NPO's compatibility with non-CFG diffusion models remains unclear."
>
> **Diffusion-NPO is independent of model architectures** and diffusion forms. It works at the prediction level instead of the model architecture level.
>
> CFG is the motivation and basis of diffusion-NPO. We cannot perform diffusion-NPO if CFG is not supported. However, please note that CFG has become almost a necessity in the diffusion field for improved generation quality.
>
> Previous works [9–12] show that even for unconditional diffusion models, we can still adopt CFG to significantly improve generation performance.

---

> ### Author Response · Authors · 2024-11-23
> **Response to Reviewer Fyov [3/4]**
>
> > **Q1** "How does NPO distinguish between truly negative attributes and those that are simply neutral or contextual?"
>
> Please refer to our reply to W1.
>
> > **Q2** "What are the computational costs of NPO, especially in terms of memory and processing time for real-time applications?"
>
> Training: Since NPO requires minor modifications to the original preference optimization methods, it requires the same computation costs.
>
> Inference: Since all the models used in the paper adopt CFG for inference, NPO requires the same computation costs for inference.  On the other hand, as illustrated in our limitations, Diffusion-NPO requires the storage of two different weight offsets for inference. However, since preference optimization can be achieved with parameter-efficient tuning techniques such as LoRA (LoRA is a low-rank approximated weight residual), it typically requires less than 1% additional parameters. For instance, we adopt the LoRA rank 8 for training, which only takes 0.17% additional parameters.
>
> Again, please kindly note that this paper focuses on improving model alignment with human preferences, not real-time applications. Due to the inherently slow iterative denoising process, diffusion models are unsuitable for real-time use under standard settings, and none of the prior works [1-8] have achieved real-time performance. Thus, real-time application is not addressed here.
>
>
> > **Q3** "How sensitive is NPO's performance to parameter settings, and what are the recommended heuristics for tuning them effectively"
>
> We follow the default training configurations of original preference optimization methods [1-3] and find those default hyperparameters work well for NPO. We also conduct a hyperparameter sensitivity analysis in lines 476-485, Fig. 9 (visual-based), and Fig. 10 (metric-based).   We tuned the parameters with a simple heuristic grid search. Specifically, we first choose parameters from 10^{k}, k= ..., -2, -1, 0, 1, 2,... to find the optimal k. Then we conduct simple grid search to find better hyperparameters.
>
> > **Q4** "How well does NPO perform on domain-specific datasets like medical imaging or abstract art?"
>
> Please refer to our reply to W4.
>
> > **Q5** "Beyond quantitative metrics, what is the perceived quality improvement from NPO based on qualitative user evaluations"
>
> Please refer to our reply to w2.
>
>
> > **Q6** "How does NPO mitigate the risk of reinforcing biases present in preference datasets"
>
> As illustrated in the paper, we adopt the pickapic dataset proposed in PickScore when training the DPO-based NPO model.
>
> On one hand, PickScore has already made efforts to minimize bias to the greatest extent possible. While it is true that some degree of bias still exists in the dataset, as acknowledged in the PickScore paper, the dataset was constructed through a comprehensive evaluation by a large and diverse population. This broad aggregation of assessments helps mitigate biases inherent in the data, ensuring it is as objective as possible within practical limits.
>
> On the other hand, our method emphasizes making effective use of the dataset, rather than aiming to completely eliminate its biases. While addressing bias is an important and ambitious goal, we believe it requires dedicated efforts in future specialized research, as it extends beyond the primary focus of our current work.
>
> In addition, note that our method is based on Stable Diffusion, which includes an important safety checker capable of detecting and filtering generated content. This allows our method to leverage it for safety checks on the generated results, further reducing potential negative biases, such as sexuality.

---

> ### Author Response · Authors · 2024-11-23
> **Response to Reviewer Fyov [4/4]**
>
> > **Q7** "Ethical guidelines and limitations on NPO's use to ensure responsible application"
>
> Thank you for the suggestion. Negative Preference Optimization (NPO) is designed to enhance model outputs by explicitly training the diffusion models to recognize and avoid generating results that are not preferred by users. By leveraging "undesirable knowledge" in a controlled manner, NPO provides a proactive mechanism for steering generative models away from harmful, inappropriate, or irrelevant outputs, which directly aligns with the principles of responsible AI.
>
> To ensure ethical and responsible deployment, we propose the following guidelines and limitations for NPO's application:
>
> Prevention of Malicious Use: NPO should strictly be restricted to contexts where the objective is to enhance user satisfaction or reduce harm. Clear safeguards must be in place to prevent its misuse for generating harmful, biased, or deceptive content, such as misinformation or discriminatory artifacts.
>
> Transparent Disclosure: The incorporation of NPO should be transparently documented, allowing stakeholders to understand its role in mitigating undesirable outputs. This is critical to build trust in its application, especially in sensitive domains.
>
> Ethical Oversight: The application of NPO must be guided by rigorous ethical review processes to evaluate potential downstream impacts, particularly when undesirable knowledge might inadvertently influence adjacent areas. Regular audits and public engagement can help ensure accountability.
>
> Focused Scope: NPO should only be applied in areas where controlling and minimizing negative outputs is essential—such as user-facing systems or domains with stringent safety requirements—to avoid overfitting to narrowly defined "undesirability" that may limit broader applicability.
>
> By embedding these practices into its development and deployment, NPO has the potential to act as a robust framework for aligning generative models with user preferences and societal values. This structured approach ensures that the technique is applied with care, precision, and an unwavering commitment to ethical AI.
>
>
>
> [1] Diffusion model alignment using direct preference optimization. CVPR 2024.
>
> [2] Aligning preference with denoising performance at each step.
>
> [3] Video Diffusion Alignment via Reward Gradients.
>
> [4] Training Diffusion Models with Reinforcement Learning.
>
> [5] Using Human Feedback to Fine-tune Diffusion Models without Any Reward Model. CVPR 2024.
>
> [6] Large-scale Reinforcement Learning for Diffusion Models. ECCV 2024.
>
> [7] Directly Fine-Tuning Diffusion Models on Differentiable Rewards. ICLR 2024.
>
> [8] DPOK: Reinforcement Learning for Fine-tuning Text-to-Image Diffusion Models. NeurIPS 2023.
>
> [9] Improving sample quality of diffusion models using self-attention guidance.  CVPR 2023.
>
> [10] Self-Rectifying Diffusion Sampling with Perturbed-Attention Guidance. ECCV 2024.
>
> [11] Guiding a Diffusion Model with a Bad Version of Itself. NeurIPS 2024 Oral.
>
> [12] Smoothed Energy Guidance: Guiding Diffusion Models with Reduced Energy Curvature of Attention. NeurIPS 2024.

---

> > ### Comment · Area_Chair_a7xu · 2024-11-24
> > **Discussion Period Ending Soon**
> >
> > Dear Reviewer,
> >
> > The discussion period will end soon. Please take a look at the author's comments and begin a discussion.
> >
> > Thanks, Your AC

---

> ### Author Response · Authors · 2024-11-25
> **Discussion deadline is approaching**
>
> Dear Reviewer Fyov,
>
> As the discussion deadline approaches, we kindly ask if our response has addressed your concerns. Please feel free to share any additional questions or feedback, and we’ll be happy to provide further clarification.
>
> Best regards,
>
> The Authors

---

> > ### Comment · Area_Chair_a7xu · 2024-12-01
> > **Discuss with the Authors**
> >
> > Dear Reviewer,
> >
> > Please discuss your reviews with the authors and consider the points they mention. Discussion and re-evaluation is a critical part of the review process.
> >
> > Thanks, Your AC

---

> > ### Comment · Reviewer_Fyov · 2024-12-02
> >
> > Thank you for your detailed and thoughtful rebuttal. I am pleased to see that the majority of my concerns have been addressed and I would like to confirm my decision to accept the paper.

---

> > > ### Author Response · Authors · 2024-12-02
> > > **Thank you sincerely for confirming the decision to accept Diffusion-NPO**
> > >
> > > Dear Reviewer Fyov,
> > >
> > > We sincerely appreciate your confirmation of acceptance and are pleased that our rebuttal has addressed most of your concerns. In light of this, we kindly request that you consider revising the rating, as it still remains at 5, which is marginally below acceptance.
> > >
> > > Best regards,
> > >
> > > The Authors

---

### Official Review · Reviewer_ka33 · 2024-11-06

**Soundness:** 3
**Presentation:** 3
**Contribution:** 3
**Rating:** 8
**Confidence:** 3

**Summary:**

This paper presents a simple and general method for aligning images generated from diffusion models with human preference. The key idea is to distinguish not only favorable images but also undesired ones. To accomplish this, the paper builds on various existing preference optimization approaches and propose negative preference optimization (NPO), which in essence reverses the order of the ranked image pairs for training a second set of diffusion weights that emphasize negative images. The experiments demonstrate strong qualitative results and favorable user study results in comparison to baselines without NPO.

**Strengths:**

- Simplicity and generality. The method is simple, intuitive, and can augment a variety of existing preference alignment approaches without additional training data and learning objectives.
- Strong results. The method achieves superior qualitative and quantitative results in comparison to baselines not using NPO.
- Efficiency. The method obtain stronger results without losing inference-time efficiency.

**Weaknesses:**

I did not find major weaknesses of the paper.

Disclaimer: While I find the approach interesting and reasonable, I am not an expert on preference optimization of diffusion models, so I am not able to comment on the novelty of the method and the selection of baselines / benchmarks.

**Questions:**

I am wondering whether it is optimal to share the same training data for PO and NPO. Humans prefer certain images for obvious reasons: higher visual quality and/or stronger semantic alignment with the text prompt. Oftentimes, the worse image in a training pair for PO still looks reasonable in quality and semantic alignment. The pool of negative images, however, can be larger and more diverse. For example, a negative image can be completely irrelevant to the text prompt, or can have arbitrary artifacts.

---

> ### Author Response · Authors · 2024-11-23
> **Thank you for your insightful and detailed review [1/1]**
>
> Thank you for your insightful and detailed review! We are delighted that you found our work to be simple, intuitive, and effective! Below, we address your questions in detail.
>
> > **Q1:**  it is optimal to share the same training data for PO and NPO?
>
> Thank you for your insightful and thought-provoking question!
>
> In our original implementation, to ensure a fair comparison, we utilized the default training data pairs, data processing methods, and configurations from the original preference optimization techniques to train NPO (as described in Ln 372). This decision was made to isolate the impact of training data distribution, thereby highlighting the simplicity and effectiveness of NPO.
>
> In response to your question regarding the impacts of better designed negative dataset, we conducted **additional experiments** to analyze the influence of training data. These experiments were performed using Diffusion-DPO-based NPO, as Diffusion-DPO is the only setting in our framework that requires training data.
>
> To explore the impact of training data, we examined two specific scenarios:
> 1. Perturbed Text Prompts:
> In this scenario, we randomly shuffled the sequence of text prompts within batches to create irrelevant image-text pairs. This led to consistent performance degradation, suggesting that even for negative preference training, maintaining proper image-text pairings is crucial.
> 2. Random Image Corruptions:
> Here, we applied random corruptions, such as Gaussian blur, to the negative images. We observed that mild corruptions improved performance. However, excessively strong corruptions caused the unconditional outputs to deviate significantly from the conditional outputs, which could lead to inference failures.
>
> These findings indicate that the image quality and image-text consistency of training data remain essential, even in the context of negative preference optimization.
>
>
> The quantitative evaluation is shown in the following table. All the win-rate are compared with the original NPO.
>
> | Method                     | AES Mean Score | AES Win-Rate | HPS Mean Score | HPS Win-Rate | ImageReward Mean Score | ImageReward Win-Rate | PickScore Mean Score | PickScore Win-Rate |
> | -------------------------- | -------------- | ------------ | -------------- | ------------ | ---------------------- | -------------------- | -------------------- | ------------------ |
> | Original NPO               | 5.7621         | -            | 27.60          | -            | 0.3102                 | -                    | 21.58                | -                  |
> | NPO with text perturbation | 5.7407         | 45%          | 27.52          | 37.6%        | 0.3148                 | 49.2%                | 21.55                | 40.4%              |
> | NPO with data corruption   | 5.7676         | 54%          | 27.63          | 51%          | 0.3222                 | 54.0%                | 21.60                | 54.4%              |
>
> Our experiments demonstrate that NPO with appropriately applied data corruption enhances overall performance. The results highlight that effective data usage and processing strategies can further boost the effectiveness of NPO, showcasing its great potential and extendability.
>
> Thank you once again for your thought-provoking question! We believe this topic warrants deeper exploration in future work.

---

### Author Response · Authors · 2024-11-23
**General response [1/3]**

We sincerely thank all reviewers for their valuable comments and suggestions. We are sincerely grateful to the reviewers for dedicating their time and effort to review our work. We are delighted to see reviewers commenting on our work with "`innovative`", "`high significance`", and "`intuitive, simple, general, and highly effective`".


In this rebuttal, we try our best to solve the concerns of reviewers. We summarize the important and common concerns in the following:


> **Comparison with training-free CFG enhancement approaches [1] [2] (Reviewer 5Mi5, 5t3C)**

Specifically, we test with the following comparisons:

1. SEG on Stable Diffusion XL  vs Naive CFG [3] on Stable Diffusion XL.
2. DPO-optimized Stable Diffusion XL as conditional predictors and original Stable Diffusion XL as the unconditional predictors vs Naive CFG on Stable Diffusion XL.
3. DPO-optimized Stable Diffusion XL as conditional predictors and NPO-optimized Stable Diffusion XL as unconditional predictors vs Naive CFG on Stable Diffusion XL.

Note that the original Stable Diffusion XL can be viewed as an earlier version of DPO-optimized Stable Diffusion XL. Therefore, we adopt 2 as the baseline of Autoguidance.

We generate all the images with the same seed and the same prompts from the test_unique split as the text prompts. We score generated images with the HPSv2, PickScore, ImageReward, and Laion Aesthetic and compare their average scores and win-rates.

The results are shown in the following table. All the win-rate are compared with original SDXL.

| Method           | AES Mean Score | AES Win-Rate | HPS Mean Score | HPS Win-Rate | ImageReward Mean Score | ImageReward Win-Rate | PickScore Mean Score | PickScore Win-Rate |
| ---------------- | -------------- | ------------ | ------------------ | ------------ | ---------------------- | -------------------- | ------------------------ | ------------------ |
| SDXL             | 6.1142         | -            | 27.89              | -            | 0.6246                 | -                    | 22.06                    | -                  |
| SEG[2]           | 6.1498         | 55.2%        | 26.26              | 10.6%        | -0.1042                | 19.4%                | 20.99                    | 9.0%               |
| AutoGuidance [1] | 5.8049         | 28.8%        | 27.21              | 27.0%        | 0.4543                 | 40.0%                | 21.44                    | 22.0%              |
| NPO (Ours)       | 6.1116         | 51.4%        | 28.78              | 81.2%        | 0.9210                 | 73.6%                | 22.69                    | 82.0%              |

We can see that SEG only achieves minor improvement on Laion-Aesthetic compared to the original baseline of SDXL. All the other metrics show that SEG significantly underperforms the naive CFG strategy with SDXL. This can be attributed to that SEG was originally proposed to tackle the circumstances where the prompt is not used. When applying AutoGuidance, we find the outputs tend to be blurry. Our quantitative evaluation also proves our claim. This can be attributed to the prediction difference between original SDXL and DPO-optimized SDXL being too large (as analyzed in lines 340-344)

We also test on the Stable Diffusion v1-5 for further comparison:

1. DPO-optimized Stable Diffusion v1-5 as conditional predictors and original Stable Diffusion v1-5 as the unconditional predictors vs Naive CFG on DPO-optimized Stable Diffusion v1-5.

2. DPO-optimized Stable Diffusion v1-5 as conditional predictors and NPO-optimized Stable Diffusion v1-5 as the unconditional predictors vs Naive CFG on DPO-optimized Stable Diffusion v1-5.

We did not test SEG on Stable Diffusion v1-5 since we did not find open-source implementation of SEG on Stable Diffusion v1-5.

The results are shown in the following table. All the win-rate are compared with dpo-optimized Stable Diffusion v1-5.

| Method           | AES Mean Score | AES Win Rate | HPS Mean Score | HPS Win Rate | ImageReward Mean Score | ImageReward Win Rate | PickScore Mean Score | PickScore Win Rate |
| ---------------- | -------------- | ------------ | ------------------ | ------------ | ---------------------- | -------------------- | ------------------------ | ------------------ |
| DPO              | 5.6484         | -            | 27.19              | -            | 0.2652                 | -                    | 21.12                    | -                  |
| AutoGuidance [1] | 5.6277         | 51.2%        | 26.96              | 34.6%        | 0.1251                 | 42.0%                | 21.26                    | 58.2%              |
| NPO              | 5.7621         | 68.8%        | 27.60              | 76.6%        | 0.3102                 | 59.2%                | 21.58                    | 84.4%              |

NPO still consistently outperforms the compared baselines.

---

> ### Author Response · Authors · 2024-11-23
> **General response [2/3]**
>
> **Difference and comparison with scaling up the existing DPO method. (Reviewer  5t3C)**
>
> The default training iteration of DPO and NPO are both 2,000 iterations. To show the effectiveness of NPO compared to solely scaling up existing DPO method. We adopt the official code of DPO and extend the training iteration of DPO to 20,000 iterations (i.e., 10x training cost). In the following table, we use the 1x, 2x, ... to denote the training iterations of each model.
>
> We summarize the quantitative evaluation results in the following table.
>
> | Method               | AES Mean | AES Win Rate | HPS Mean | HPS Win Rate | ImageReward Mean | ImageReward Win Rate | PickScore Mean | PickScore Win Rate |
> | -------------------- | -------- | ------------ | -------- | ------------ | ---------------- | -------------------- | -------------- | ------------------ |
> | DPO (1x)             | 5.6320   | 32.2%        | 27.17    | 34.0%        | 0.2576           | 42.6%                | 21.00          | 18.6%              |
> | DPO (1x) + NPO (1x)  | 5.7667   | 67.8%        | 27.46    | 66.0%        | 0.3090           | 57.4%                | 21.47          | 81.4%              |
> | DPO (2x)             | 5.6178   | 30.8%        | 27.21    | 32.2%        | 0.2674           | 41.8%                | 21.06          | 19.6%              |
> | DPO (2x) + NPO (1x)  | 5.7662   | 69.2%        | 27.51    | 67.8%        | 0.3239           | 58.2%                | 21.50          | 80.4%              |
> | DPO (3x)             | 5.6214   | 27.8%        | 27.28    | 33.4%        | 0.3322           | 49.2%                | 21.08          | 20.6%              |
> | DPO (3x) + NPO (1x)  | 5.7685   | 72.2%        | 27.60    | 66.6%        | 0.3362           | 50.8%                | 21.54          | 79.4%              |
> | DPO (4x)             | 5.6399   | 31.8%        | 27.36    | 33.2%        | 0.3574           | 46.0%                | 21.12          | 21.6%              |
> | DPO (4x) + NPO (1x)  | 5.7767   | 68.2%        | 27.60    | 66.8%        | 0.3612           | 54.0%                | 21.55          | 78.4%              |
> | DPO (5x)             | 5.6718   | 33.4%        | 27.35    | 31.2%        | 0.3540           | 44.4%                | 21.15          | 18.6%              |
> | DPO (5x) + NPO (1x)  | 5.7868   | 66.6%        | 27.65    | 68.8%        | 0.3688           | 55.6%                | 21.59          | 81.4%              |
> | DPO (6x)             | 5.6632   | 34.6%        | 27.35    | 34.6%        | 0.3664           | 45.8%                | 21.18          | 20.6%              |
> | DPO (6x) + NPO (1x)  | 5.7685   | 65.4%        | 27.63    | 65.4%        | 0.3780           | 54.2%                | 21.59          | 79.4%              |
> | DPO (7x)             | 5.6648   | 36.4%        | 27.37    | 33.0%        | 0.3980           | 45.0%                | 21.19          | 21.0%              |
> | DPO (7x) + NPO (1x)  | 5.7576   | 63.6%        | 27.65    | 67.0%        | 0.4141           | 55.0%                | 21.59          | 79.0%              |
> | DPO (8x)             | 5.6605   | 37.4%        | 27.32    | 31.4%        | 0.3840           | 49.8%                | 21.18          | 19.4%              |
> | DPO (8x) + NPO (1x)  | 5.7544   | 62.6%        | 27.62    | 68.6%        | 0.4122           | 50.2%                | 21.58          | 80.6%              |
> | DPO (9x)             | 5.6438   | 37.6%        | 27.22    | 27.2%        | 0.3679           | 44.8%                | 21.10          | 20.8%              |
> | DPO (9x) + NPO (1x)  | 5.7463   | 62.4%        | 27.61    | 72.8%        | 0.4121           | 55.2%                | 21.56          | 79.2%              |
> | DPO (10x)            | 5.6264   | 39.6%        | 27.10    | 28.8%        | 0.3214           | 42.0%                | 21.05          | 19.6%              |
> | DPO (10x) + NPO (1x) | 5.7284   | 60.4%        | 27.51    | 71.2%        | 0.3986           | 58.0%                | 21.51          | 80.4%              |
>
> We can see from the table that:
>
> 1. No matter how long DPO is trained. The NPO weight offset trained with only 1x training cost can consistently and significantly improve the overall performance.
> 2. DPO (1x) + NPO (1x) achieves Aesthetic Score (5.7667), HPS Score (27.46), PickScore (21.47). The performance significantly outperforms the DPO (10x), which requires 10/(1+1) = 5 times longer training cost with Aesthetic Score (5.6264), HPS Score (27.10), PickScore (21.05)
>
> **We believe this is very strong evidence of the effectiveness of diffusion-NPO.**

---

> > ### Author Response · Authors · 2024-11-23
> > **General response [3/3]**
> >
> > > **Can the data for NPO training to be more corrupted to improve performance? (Reviewer ka33)**
> >
> > we conduct additional experiments to investigate the influence of training data. All the additional experiments are based on Diffusion-DPO considering its simplicity and it is the only setting that requires data for training.
> >
> > To investigate the influence, we investigate two settings:
> >
> > 1) Text prompts. In this setting, we randomly pertubate the sequence of text prompts within batches to construct irrelevant image-text pairs. However, we observe consistent performance degradation in this setting. This might indicate that even for negative preference training, proper image-text pairs are still important.
> > 2) In this setup, we apply random corruptions, such as GaussianBlur, to the negative images. Our findings indicate that mild corruptions lead to performance improvements. However, overly strong corruptions can cause the unconditional outputs to deviate excessively from the conditional outputs, potentially resulting in inference failure.
> >
> > The quantitative evaluation is shown in the following table. All the win-rate are compared with the original NPO.
> >
> > | Method                     | AES Mean Score | AES Win-Rate | HPS Mean Score | HPS Win-Rate | ImageReward Mean Score | ImageReward Win-Rate | PickScore Mean Score | PickScore Win-Rate |
> > | -------------------------- | -------------- | ------------ | -------------- | ------------ | ---------------------- | -------------------- | -------------------- | ------------------ |
> > | Original NPO               | 5.7621         | -            | 27.60          | -            | 0.3102                 | -                    | 21.58                | -                  |
> > | NPO with text perturbation | 5.7407         | 45%          | 27.52          | 37.6%        | 0.3148                 | 49.2%                | 21.55                | 40.4%              |
> > | NPO with data corruption   | 5.7676         | 54%          | 27.63          | 51%          | 0.3222                 | 54.0%                | 21.60                | 54.4%              |
> >
> > We can observe that NPO with proper data corruption indeed improves the overall performance.  The experimental results show that effective strategies for data usage and processing can further improve the performance of NPO, showing the great potential and extendable property of NPO. Thank you for the intriguing question.
> >
> > > **While the paper is innovative, the simple reversal of preference pair rankings may oversimplify the complexity of human aesthetics. (Reviewer Fyov.)**
> >
> >
> >
> > We agree that the simple reversal of preference rankings may not capture the full complexity of human aesthetics, which are inherently subjective and multifaceted. However, our approach serves as a deliberate simplification to address a specific problem: steering the model away from generating outputs that are misaligned with user preferences. By reversing the rankings, we focus on penalizing undesired outputs effectively, which allows us to reduce the likelihood of such results without significantly increasing computational complexity. **We view our method as a foundational step rather than a comprehensive solution.** It demonstrates the potential of using preference data in novel ways, but we acknowledge that more sophisticated approaches could account for the nuances of human aesthetics more accurately. Future work could involve incorporating richer datasets, leveraging advanced ranking models, or even introducing mechanisms to capture context-dependent aesthetic judgments. This could enhance the ability of the model to generate outputs that align more closely with the complexity of human preferences.
> >
> >
> > Please refer to the following rebuttals for other specific concerns and more details. **We are looking forward to your further reply and discussion.**
> >
> > Sincerely,
> >
> >
> >
> > Authors
> >
> > ---
> >
> >
> >
> > [1] Guiding a Diffusion Model with a Bad Version of Itself. NeurIPS Oral.
> >
> > [2] Smoothed Energy Guidance: Guiding Diffusion Models with Reduced Energy Curvature of Attention. NeurIPS 2024.
> >
> > [3] Classifier-Free Diffusion Guidance.

---

### Meta-Review · Area_Chair_a7xu · 2024-12-20

**Metareview:**

The paper addresses the problem of preference optimization for diffusion model and identifies an overlooked component: negative preference. The paper proposes to use standard PO techniques but specifically train the diffusion model to be able to control how far the sample is pushed against negative preference (defined as the negation of the normal reward model). Then at inference time, the model integrates this negative preference in the standard CFG formulation as part of the normally unconditional generation. Specifically a merged weight baseline is proposed where the DPO is used for the positive signal and a merged model of DPO and original for the negative signal. The paper then shows that the merged model performs better than a standard DPO model. The paper shows that the merged model performs better than the standard one. The paper also shows that this can improve text to video as well.

The reviewers were mostly positive on the paper mentioning few weaknesses. The reviewers mentioned that they found the paper innovative, simple, and effective.

I agree in general with the reviewers and advocate for acceptance of the paper. I find that the strengths of the paper (the simplicity and effectiveness as well as the very good presentation) outweigh the weaknesses (could be more thorough on the related works, could have done more exploration of the "negative" formulation – e.g. I think exploring things like corruptions or a different signal than the negation of the reward model would have been more interesting).

As a quick side-note, there have been some explorations into using "bad" images as a negative signal. For example, I remember this post from 2023 (https://minimaxir.com/2023/08/stable-diffusion-xl-wrong/). Additionally there is actually a fairly rich literature of work exploring this exact topic (explicitly modeling negative preference) for language. However, I do not think that the authors should be penalized for work in the general public or different fields. I think in the interest of academic honesty / transparency, I would recommend including them on a website in a section named "other work" or "similar ideas", but since they're not published and/or in different fields, I don't think this is 100% required.

Note: THIS IS REQUIRED. The bottom right example of Figure 11 "preteen girls with no underware..." and the top left of Figure 15 "boy climbing into oven, lewd" MUST BE REPLACED for the paper to be accepted.

**Additional Comments On Reviewer Discussion:**

The reviewers were mostly positive on the paper. Reviewer Fyov highlighted that the paper could have been explored in additional fields or contexts. Reviewer 5Mi5 requested comparisons against two conceptually related papers that the authors provided.

---

### Decision · Program_Chairs · 2025-01-22

Accept (Poster)